# Formation of Reactive Nitrogen Species Promoted by Iron Ions through the Photochemistry of a Neonicotinoid Insecticide

Zhu Ran[1,2], Yanan Hu[1,2,3*], Yuanzhe Li[1,2], Xiaoya Gao[1,2], Can Ye[4], Shuai Li[5], Xiao Lu[5], Yongming Luo[1,2,3], Sasho Gligorovski [6,7,8*], Jiangping Liu[1,2*]

[1]Faculty of Environmental Science and Engineering, Kunming University of Science and Technology, Kunming 650500, China;
[2]The Innovation Team for Volatile Organic Compounds Pollutants Control and Resource Utilization of Yunnan Province, The Higher Educational Key Laboratory for Odorous Volatile Organic Compounds Pollutants Control of Yunnan Province, Kunming 650500, China;
[3]Faculty of Chemical Engineering, Kunming University of Science and Technology, Kunming 650500, China;
[4]Faculty of Environmental Science and Engineering, Peking University, Beijing 100871, China;
[5]School of Atmospheric Sciences, Sun Yat-sen University, Southern Marine Science and Engineering Guangdong Laboratory (Zhuhai), Zhuhai, Guangdong 519082, China;
[6]State Key Laboratory of Organic Geochemistry and Guangdong Provincial Key Laboratory of Environmental Protection and Resources Utilization, Guangzhou Institute of Geochemistry, Chinese Academy of Sciences, Guangzhou 510 640, China
[7]Guangdong-Hong Kong-Macao Joint Laboratory for Environmental Pollution and
Control, Guangzhou Institute of Geochemistry, Chinese Academy of Science, Guangzhou 510640, China;
[8]Chinese Academy of Science, Center for Excellence in Deep Earth Science, Guangzhou, 510640, China.

*Correspondence to*: gligorovski@gig.ac.cn; liujiangping18@mails.ucas.ac.cn; huyanan0917@163.com

**Abstract.** Nitrous acid (HONO) and nitrogen oxides (NOx = NO + NO$_2$) are important atmospheric pollutants and key intermediates in the global nitrogen cycle, but their sources and formation mechanisms are still poorly understood. Here, we investigated the effect of soluble iron (Fe$^{3+}$) on the photochemical behaviour of a widely used neonicotinoid (NN) insecticide, nitenpyram (NPM), in the aqueous phase. The yields of HONO and NOx increased significantly when NPM solution was irradiated in the presence of iron ions (Fe$^{3+}$). We propose that the enhanced HONO and NO$_2$ emissions from the photodegradation of NPM in the presence of iron ions result from the redox cycle between Fe$^{3+}$ and Fe$^{2+}$ and the generated reactive oxygen species (ROS) by the electron transfer between the excited triplet state of NPM and the molecular oxygen (O$_2$). Using the laboratory-derived parametrization based on kinetic data and gridded downward solar radiation, we estimate that the photochemistry of NPM induced by Fe$^{3+}$ releases 0.50 and 0.77 Tg N year$^{-1}$ of NOx and HONO to the atmosphere, respectively.

This study suggests a novel source of HONO and NOx during daytime and potentially helps to narrow the gap between the field observations and model outcomes of HONO in the atmosphere. The suggested photochemistry of NPM can be an important contribution to the global nitrogen cycle affecting the atmospheric oxidizing capacity as well as the climate change.

## 1 Introduction

Neonicotinoids (NNs) are a class of systemic insecticides that have been widely used in agriculture and horticulture since the 1990s (Bass et al., 2015) accounting for one-third of the total world insecticide market (Simon et al., 2015) with growing use in the past decades (Botías et al., 2015; Morrissey et al., 2015). They are highly water-soluble and persistent in the environment, and can be transported to surface waters via runoff, leaching, or spray drift. NNs have been detected in various aquatic ecosystems, such as rivers, lakes, wetlands, and coastal waters, at concentrations ranging from 12.45 ng L$^{-1}$ to 225 μg L$^{-1}$ (Pan et al., 2020; Anderson et al., 2013). Increasing public perception of NNs insecticides pollution, led to significant research efforts devoted to revealing the effect of insecticide application on human (Cimino et al., 2017; Han et al., 2018), birds (Hallmann et al., 2014; Millot et al., 2017) , animals (Morrissey et al., 2015; Gibbons et al., 2015) and pollinators (especially bees) (Kessler et al., 2015; Raine et al., 2015; Goulson et al., 2015). In the environment, NNs insecticides can undergo various chemical processes, photolysis being one of the major fate (Lu et al., 2015; González-Mariño et al., 2018). Recent studies have focused mainly on photochemistry of NNs insecticides and related atmospheric lifetimes, and quantum yields (Lu et al., 2015; González-Mariño et al., 2018; Aregahegn et al., 2017; Aregahegn et al., 2018). It has been shown that the ozonolysis of NNs insecticides on various surfaces could contribute to the formation of gaseous nitrous oxide nitrous acid (HONO) (Wang et al., 2020). Gaseous nitrous oxide (N$_2$O), which is a potent greenhouse gas, was previously identified as the gas-phase product in the photolysis of solid thin films of NNs (nitenpyram, acetamiprid, thiamethoxam, thiacloprid, clothianidin and dinotefuran), with yields of ΔN$_2$O/ΔNN > 0.5 in air at both 313 and 254 nm (Wang et al., 2019; Aregahegn et al., 2017; Aregahegn et al., 2018). Palma et al (Palma et al., 2020) used a gas-flow reactor connecting with a NOx analyzer, and the production of gaseous NO/NO$_2$ was founded during irradiation (300-450 nm) of

imidacloprid. However, the crucial role of the NNs insecticides in the global nitrogen cycle at the air-water interface is largely unknown.

Nitenpyram (NPM) is one of the most commonly used NNs insecticides. It represents a systemic NNs insecticide which is widely distributed among soil, dust particles and in the aqueous environment (Botías et al., 2015; Ezell et al., 2019). Once released in the environment, NPM will be transformed into other products by absorbing sunlight ($\lambda>290$ nm) and/or reacting with atmospheric oxidants such as the hydroxyl radical (OH) and ozone ($O_3$) (Wang et al., 2020). The NPM is a nitroalkene, which is structurally similar to nitroaromatic compounds (Ar-$NO_2$). Previous studies have indicated that the photolysis of Ar-

$NO_2$ can be a source of HONO and NOx in the atmosphere (Fukuhara et al., 2006; Yang et al., 2001; Bejan et al., 2021). HONO represents one of the main sources of OH radicals in the urban atmosphere contributing by up to 80% of the total OH production ( Alicke et al., 2003; Young et al., 2012; Zheng et al., 2020). The main identified HONO sources in the urban air are the photolysis of nitrates (Ye et al., 2017; Gen et al., 2021) and light-induced heterogeneous reaction of $NO_2$ with environmental surfaces (Liu et al., 2019; Liu et al., 2020; Liu et al., 2023; Monge et al., 2010; Han et al., 2016 ). Yet, there is

a discrepancy between the modeled HONO values and field observations of HONO during the daytime, suggesting that there are missing HONO sources in the atmosphere. Meanwhile, the quantification of NOx is also of great significance for the atmospheric cycle of nitrogen species as NOx plays a crucial role in the photochemical smog and acid rain formation. Therefore, it is worthwhile to explore the contribution of NPM photolysis to HONO and NOx, which in turn can offer a guidance for the development of more sustainable next generation insecticide products.

Iron species are ubiquitous on earth surfaces, including water, soil and air-water interface (Gen et al., 2021). Recent study (Kebede et al., 2016) showed that one of less explored HONO sources could be highly dependent on the photochemical reaction of iron. The photosensitivity, oxidation state and catalytic properties of iron could enable it to possibly react with NNs insecticides compounds which are enriched at air-water interface. Previous studies on the mechanism of NNs oxidation in the ferric aqueous phase have focused on the photo-Fenton reaction (Malato et al., 2021; Lacson et al., 2018; Wang et al.,

2022; Nguyen et al., 2020; Sedaghat et al., 2016) and heterogeneous-phase photocatalysis (Rózsa et al., 2019; Sun et al., 2019; Hayat., 2019; Soltani-nezhad., 2019). As recently reported, the photolysis of iron can generate several reactive oxygen species (ROS) such as $O_2\bullet^-/HO_2\bullet$, which can trigger the redox cycle between $Fe^{3+}$ and $Fe^{2+}$ (Gen et al., 2021) as well as promote NNs insecticides oxidation. Meanwhile, iron ions inhibit the degradation of organic matter through the formation of complexes, mainly due to fluorescence bursting. The complexation may cause the inhibition of the excited singlet state and

thus the photoformation of the triple excited state (Wan et al., 2019). In addition to the NN insecticides and iron photosensitizers, the nitrate ($NO_3^-$) and nitrite ($NO_2^-$) can also absorb sunlight in the actinic region and initiate production of ROS (Vione et al., 2019). Moreover, the reaction between $Fe^{2+}$ and $NO_3^-$ may be a potentially important source of HONO (Gen et al., 2021). To this end, we suggest that the photolysis of NPM in the presence of iron may contribute to a missing atmospheric HONO source.

To our knowledge, this is a first investigation to measure the photochemical production of HONO and NOx from NPM photolysis in absence and in the presence of soluble iron. The photolysis frequency of HONO ($J_{NPM \rightarrow HONO}$), NO$_2$ ($J_{NPM \rightarrow NO2}$) and NO ($J_{NPM \rightarrow NO}$) during the NPM reaction at the air-water interface was investigated. The kinetics and mechanism of HONO and NOx formation in the presence of soluble iron were evaluated. This study highlights an overlooked source of HONO and NOx from NNs-covered water surfaces which may play a critical role in atmospheric nitrogen cycle and evaluation of the atmospheric oxidation capacity.

## 2 Experimental

### 2.1 Materials and Sample Preparation

Solid NPM (Aladdin, China) was dissolved in ultra-pure water to prepare an aqueous NPM solution (0.5 mg mL$^{-1}$) before each experiment. FeCl$_3$ (98%, Aladdin China) was used as the source of different concentrations of aqueous Fe$^{3+}$ (0.1-0.8 mg mL$^{-1}$), and their solutions were prepared by dissolving the corresponding mass of FeCl$_3$ in ultra-pure water.

### 2.2 Experimental Setup

The circular reactor consisted of a double layer of quartz glass (3.4 cm height, 7.5 cm inner diameter) connected to a thermostatic bath (XOSC-20, China), which allowed operation at a constant temperature of 298 K (Figure S1). The previously prepared sample solution was placed in the circular reactor and exposed to a Xenon lamp (Perfect Light, PLS-SXE 300, China) vertically above the reactor. The Xenon lamp was 12 cm away from the liquid level of NPM. The spectral irradiance of the Xenon lamp was measured by a calibrated spectroradiometer (HP 350 UVP, China) (Figure S1). Dry air collected from an air generator (HY-3, China) was used for the experiment. During the whole experiment, a constant flow of 800 mL min$^{-1}$ of dry air was controlled by an electronic soap film flowmeter (SCal Plus, China). The UV absorption spectra of the NPM aqueous solutions in the absence and in the presence of iron ions were measured by the UV-vis double-beam spectrophotometer (Shimadzu 2600, Japan) (Figure S2, Text S1).

### 2.3 NOx, HONO, NPM and ROS measurements

NO, NO$_2$ and HONO were detected using a chemiluminescence NOx analyzer (42i, THERMO) with a molybdenum converter. NO was measured by reacting NO with O$_3$ to produce characteristic luminescence, and the intensity of luminescence was proportional to the concentration of NO. In the detection of NO$_2$, a molybdenum catalyst was used to convert NO$_2$ to NO. A quartz tube (25 cm length, 2.9 cm inner diameter) filled with a certain amount of crystalline Na$_2$CO$_3$ was introduced to capture HONO between the circular reactor outlet and the NOx analyzer. It is well known that almost all HONO molecules can contact Na$_2$CO$_3$ when using molybdenum converters, achieving high capture efficiency of HONO. Therefore, HONO can be indirectly quantified by the difference between the NO$_2$ signal and the Na$_2$CO$_3$ tube (Monge et al., 2010; Cazoir et al., 2014; Brigante et al., 2008; Zhou et al., 2018). The quantification of NPM before and after the reaction

was determined by High performance liquid chromatography (HPLC). The mobile phase was a mixture of water and acetonitrile with a flow rate of 0.5 mL min$^{-1}$ at 80:20 (v/v). The column temperature was kept at 30°C, the injection volume was 20 µL, and the detection wavelength was set to 270 nm. The external standard method was used for the quantitative determination of NPM. Photoproductions of $O_2^-\bullet$, $^1O_2$ and $\bullet OH$ (ROS) were quantified using DMPO, TEMP and DMPO as chemical probe molecules, respectively.

## 2.4 Kinetic Analysis

The NPM photolysis kinetics was described using a first-order reaction (Eq.(1)), and the half-life ($t_{1/2}$) was calculated using Eq.(2).

$$C_t = C_0 \times e^{-kt} \tag{1}$$

$$t_{1/2} = \ln(2)/k \tag{2}$$

where $C_0$ (mg ml$^{-1}$) is initial concentration of NPM, $C_t$ (mg ml$^{-1}$) is the NPM concentration at time t, and k is the first-order rate constant.

## 2.5 The photolysis frequency

The photolysis frequencies of NPM to HONO and NOx were calculated by Eq.(3) and Eq.(4), respectively.

$$J_{NPM \to HONO} = \frac{QM_{NPM} \int_0^t C_t^{HONO} dt}{60 \times 10^{-3} N_A \times t \times (m_0 + m_t)/2} \tag{3}$$

$$J_{NPM \to NOx} = \frac{QM_{NPM} \int_0^t C_t^{NOx} dt}{60 \times 10^{-3} N_A \times t \times (m_0 + m_t)/2} \tag{4}$$

Where Q (mL min$^{-1}$) and MNPM (g mol$^{-1}$) are the total flow gas rates in the reactor and the molar mass of NPM, respectively; t (min) is the irradiation time; $C_t^{NOx}$ (molecules cm$^{-3}$) is the concentration of gaseous HONO or NOx formed by photolysis of NPM during the irradiation period; $N_A$ is the Avogadro number; $M_0$ (mg) and $M_t$ (mg) are the masses at the beginning and end of the NPM photolysis experiments.

## 2.6 Flux densities of HONO and NOx

The flux densities of HONO and NOx were estimated by using the following equations:

$$HONO_{flux} = \frac{[HONO] \cdot V}{s \cdot t} \tag{5}$$

$$NOx_{flux} = \frac{[NOx] \cdot V}{s \cdot t} \tag{6}$$

where HONO flux is expressed in molecules $cm^{-2}$ $s^{-1}$, [HONO] is the concentration of HONO in molecules $cm^{-3}$, V ($cm^3$) is the volume of the reactor, S ($cm^2$) is the surface of the reactor, and t (s) is the residence time of HONO in the circular reactor.

## 2.7 Global simulation of NOx and HONO fluxes

We estimated the global inventory of the NOx and HONO fluxes produced by NPM photochemistry using the observation-constrained parametrization scheme and hourly solar radiation data. Gridded and hourly downward solar radiation data are obtained from the Modern-Era Retrospective analysis for Research and Application version 2 (MERRA-2) assimilated meteorological fields. We calculated the flux of NOx and HONO for each model grid at a horizontal resolution of $0.5° \times 0.625°$ (consistent with MERRA2 radiation dataset) following Eq-S1, Eq-S2 and Eq-S3, but assuming that the environmental NPM concentration is three orders smaller than the experimental conditions of 0.5 mg $L^{-1}$. The parameterization of HONO and NOx productions from NPM photolysis at $Fe^{3+}$ concentration of 0.025 mg $L^{-1}$ used in our estimation is based on Eq-S1, Eq-S2 and Eq-S3, and more details can be seen in the Text S2.

## 3 Results and Discussion

### 3.1 Absorbance of NPM in the presence of $Fe^{3+}$

Figure S1 shows the absorbance of NPM (0.05 mg $ml^{-1}$) in the dilute aqueous phase and at different $Fe^{3+}$ concentrations, adjusted by $FeCl_3$ along with the emission spectrum of the solar simulator and the sunlight. The presence of $Fe^{3+}$ at various initial concentrations slightly enhanced the absorbance of NPM, especially at high $Fe^{3+}$ concentration (0.08 mg $ml^{-1}$), indicating that no $Fe^{3+}$-NPM complexes were generated (Liu et al., 2020). Indeed, pH is a sensitive parameter that can significantly affect the light-absorbing properties and the degree of photochemical degradation of organic compounds (Cai et al., 2018; Zhou et al., 2019). The interaction between $Fe^{3+}$ and organics as well as possible aggregation of organics at low pH may also influence the light absorption at low wavelengths (Weishaar et al., 2003). The change of $Fe^{3+}$ concentrations may alter the pH of the system, which in turn may affect the protonation/deprotonation degree of NPM, and therefore affects its absorption spectrum (Zhou et al., 2019). The pH value of the NPM solution in the presence of $Fe^{3+}$ varies between 2.4 and 3.4, and under this pH conditions, NPM (pKa=3.1) exists in both, ionic and neutral form (Hậu et al., 2021; Bonmatin et al., 2015).

### 3.2 Kinetic analysis

Iron ions are ubiquitous in natural waters with concentrations ranging between $10^{-7}$ and $10^{-4}$ M, and even higher in contaminated waters (Li et al., 2018; Faust et al., 1990). Previous studies have shown that iron ions play an important role in the photolysis of pesticides and may affect the photodegradation of organic pollutants (Faust et al., 1990; Zhao et al., 2014). The photolysis kinetics of NPM were performed to account the loss of NPM. The photolysis of NPM at different

concentrations of $Fe^{3+}$ obeyed pseudo-first-order kinetics (Figure 1), with half-lives ranging from 135.1 to 223.6 minutes as the $Fe^{3+}$ concentration increased from 0 to 0.8 mg ml$^{-1}$ (Table S1).

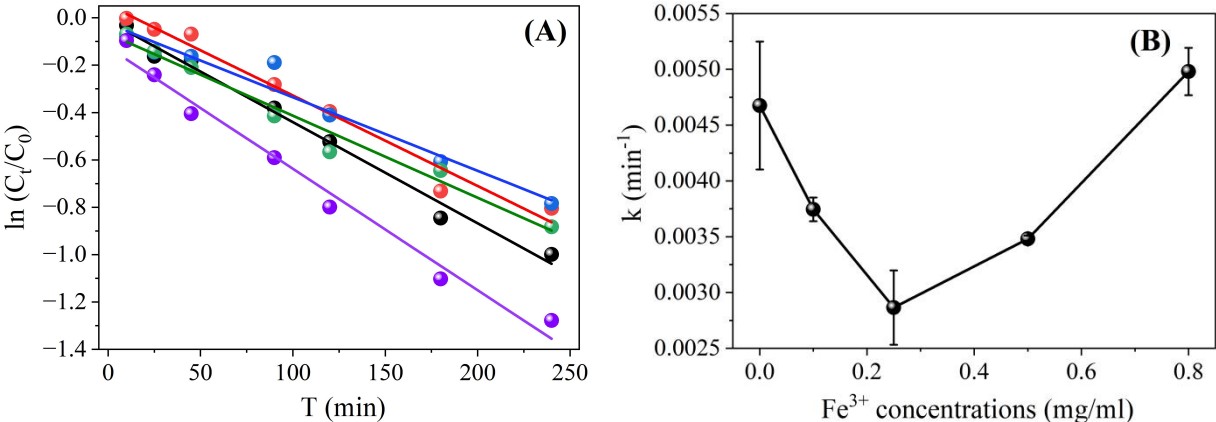

**Figure 1. (A) The kinetics of NPM (0.5 mg ml$^{-1}$) in the absence of $Fe^{3+}$ (dark line), and in the presence of different $Fe^{3+}$ concentrations: 0.1 mg ml$^{-1}$ (red line), 0.25 mg ml$^{-1}$ (blue line), 0.5 mg ml$^{-1}$ (green line) and 0.8 mg ml$^{-1}$ (purple line). (B) The rate constants of NPM light-induced degradation (0.5 mg ml$^{-1}$) at different $Fe^{3+}$ concentrations.**

The light-induced degradation of NPM was significantly inhibited at low $Fe^{3+}$ concentration ($C_{(Fe3+)}$ <0.5 mg ml$^{-1}$, Figure 1 and Table S1). In contrast, when the concentration of $Fe^{3+}$ reached 0.8 mg ml$^{-1}$, the degradation of NPM is promoted (Figure 1), exhibiting a rate constant of 0.00513 min$^{-1}$ (Table S1). Previous studies have demonstrated that the degradation of organic compounds in the presence of $Fe^{3+}$ is dose dependent (Lin et al., 2019; Deguillaume et al., 2005). For instance, $Fe^{3+}$ slightly inhibits the photodegradation of fluazaindolizine at concentration of 1-5 mg L$^{-1}$ but promotes its degradation rate at concentrations ranging between 0.1 and 0.5 mg L$^{-1}$ (Lin et al., 2019). Fang et al. (Deguillaume et al., 2005) reported that photodegradation of flupyradifurone, a novel neonicotinoid pesticide, was faster at lower $Fe^{3+}$ concentrations and slowed down with the increase of $Fe^{3+}$ concentration (Deguillaume et al., 2005).

The main reason for the inhibition effect of $Fe^{3+}$ is the attenuation of radiation due to the absorption by $Fe^{3+}$ (light screening), which reduces the light absorbance by NPM and its photodegradation. At the same time, it has been extensively confirmed that $[Fe^{3+}(OH)]^{2+}$ is the main form of $Fe^{3+}$ and exhibits great photoactivity in aqueous solution at pH=3 (Bai et al., 2023; Li et al., 2023). In the presence of $[Fe^{3+}(OH)]^{2+}$, strong oxidizing reactive oxygen species (ROS) are produced, which promote hydroxylation and degradation of NPM (Andrianirinaharivelo et al., 1995; Mazellier et al., 1997). As a result, at pH=3, the photodegradation of NPM is accelerated at high $Fe^{3+}$ concentrations.

In this study, high $Fe^{3+}$ concentration (0.8 mg ml$^{-1}$) promoted the photodegradation of NPM, and the formation of HONO and NOx (see the section below). The enhanced formation of HONO and NOx can be ascribed to ROS as described in the section 3.5.

## 3.3 HONO and NOx Formations by NPM Photolysis

The experiments of NPM photodegradation in the aqueous phase were performed to measure the HONO and NOx production in the presence of different $Fe^{3+}$ concentrations. The HONO and NOx production by spontaneous reaction of NPM in dark were negligible (Figure S3). When the NPM samples were exposed to light irradiation the concentrations of HONO and NOx quickly increased (Figure 2A).

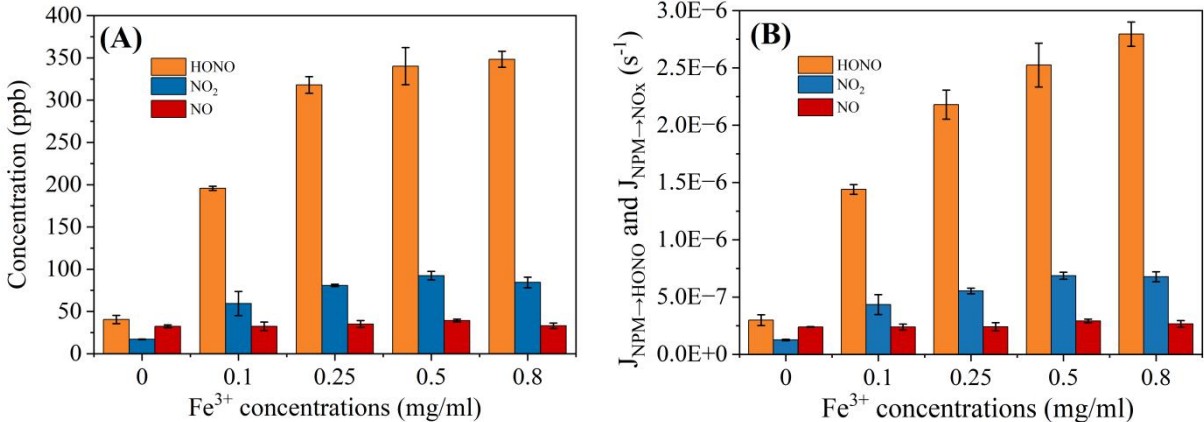

**Figure 2. (A) The concentration of NO, NO$_2$ and HONO produced by NPM photolysis at different concentrations of $Fe^{3+}$. (B) $J_{NPM \to HONO}$ and $J_{NPM \to NOx}$ from NPM at different concentrations of $Fe^{3+}$. Irradiation intensity of 169.4 W m$^{-2}$ at 300<λ<400 nm, T=298 K.**

Only the concentration of NO formed upon irradiation of NPM is almost the same in the absence of $Fe^{3+}$ and in the presence of 0.25 mg mL$^{-1}$ of $Fe^{3+}$ (Figure 2A). In the meantime, the NO$_2$ formation increased significantly with the increase of $Fe^{3+}$ concentrations and remained almost steady during the whole light exposure time (Figure S3). Moreover, when the experiments were shifted to high concentration of soluble iron (0.25-0.8 mg mL$^{-1}$), significantly enhanced NO$_2$ and NO formation were observed, and then slowly decreased with the light exposure time. In order to better understand the effect of iron on HONO and NOx production, the quantities of HONO and NOx were compared when the NPM photolysis reached a relatively stable stage (120 min). It is important to note that, the formed HONO (341 ppb) was significantly higher at the iron concentration of 0.8 mg mL$^{-1}$ compared to the HONO (37 ppb) formed in the absence of iron. Similarly, the quantity of the formed NO$_2$ increased from 17 ppb in the absence of iron to 96 ppb in the presence of 0.5 mg mL$^{-1}$ of $Fe^{3+}$. However, further increase of the iron concentration to 0.8 mg mL$^{-1}$ tend to decline the production of NO$_2$. Figure 2 shows that the NO concentrations almost remained unchanged with the increase of iron concentration. To quantify the photolysis quantum yields of HONO, NO$_2$ and NO formation from NPM photolysis, we estimated the photolysis frequency of HONO ($J_{NPM \to HONO}$), NO$_2$ ($J_{NPM \to NO2}$) and NO ($J_{NPM \to NO}$) formation, respectively (Figure 2B). $J_{NPM \to HONO}$ varied from $(2.99\pm0.46)\times10^{-7}$ s$^{-1}$ in the absence of $Fe^{3+}$ to $(2.79\pm0.10)\times10^{-6}$ s$^{-1}$ in the presence of 0.8 mg ml$^{-1}$ $Fe^{3+}$. Simultaneously, $J_{NPM \to NO2}$ increased ca. 5-fold from $(1.25\pm0.06)\times10^{-7}$ s$^{-1}$ in the absence of $Fe^{3+}$ to $(6.77\pm0.44)\times10^{-7}$ s$^{-1}$ at 0.8 mg ml$^{-1}$ $Fe^{3+}$. Regarding the $J_{NPM \to NO}$, there was nearly no discernible changes observed, with values ranging from $(2.38\pm0.27)\times10^{-7}$ s$^{-1}$ to $(2.92\pm0.15)\times10^{-7}$ s$^{-1}$. Previous studies (Yang et al., 2021) have found that the photolysis frequency of HONO and NO in nitrophenol solid-phase films (4-

nitrophenol, 4-nitrocatechol, 3,5-dinitrosalicylic acid, 3-nitrosalicylic acid, and 5-nitrosalicylic acid) varied in the ranges of $(0.34\text{-}4.16)\times10^{-7}$ and $(0.38\text{-}3.21)\times10^{-7}$ $s^{-1}$, respectively, when irradiated by xenon lamps. NPM liquid-phase photolysis produced HONO and NOx at the photolysis frequency of $10^{-7}$, but the addition of iron resulted in the photolysis frequency of $10^{-6}$ for HONO, suggesting that iron significantly facilitated the release of HONO. In order to compare the efficiency of NPM at different $Fe^{3+}$ concentrations in producing HONO and NO, $\Phi_{HONO}$ and $\Phi_{NOx}$ were displayed (Table S2). It can be concluded that NPM with high $Fe^{3+}$ concentrations had more important HONO formations as compared to pure NPM.

### 3.4 HONO and NOx Surface Flux Densities

Figure 3 summarizes the results obtained in terms of HONO formation rates per unit of exposed surface area, flux densities of HONO, $NO_2$, and NO.

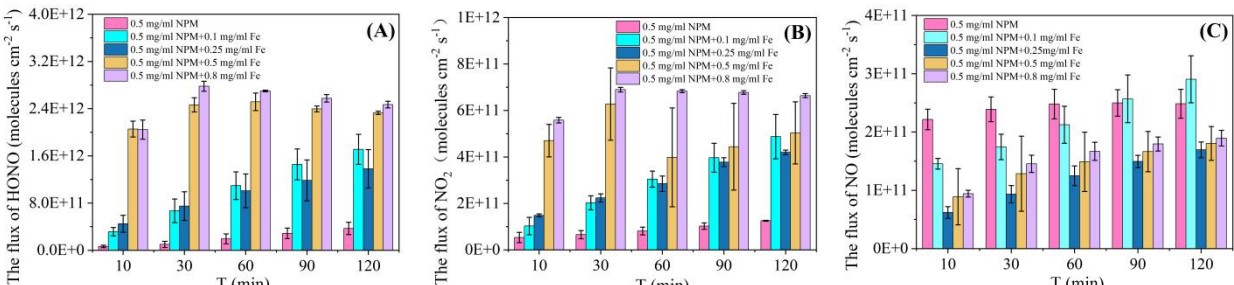

**Figure 3. Flux densities of HONO (A), $NO_2$ (B) and NO (C) determined in function of photolysis time of NPM in the presence of different concentrations of $Fe^{3+}$.**

The flux densities values of HONO and NOx indicate that direct photolysis dominated the transformation process of the NPM samples in the absence of $Fe^{3+}$. However, the introduction of soluble iron, leads to significantly increased HONO and $NO_2$ yields during the first 10 min reaction time. The further progress of the reaction up to two hours leads to slightly increased flux densities of $NO_2$ and HONO. In contrast, the NO formation showed a slow decrease after the addition of $Fe^{3+}$. A recent study (Aregahegn et al., 2017) demonstrated that photolysis of solid film consisting of imidacloprid (IMD) did not lead to HONO and NOx formation, but $N_2O$ was rather the main gas-phase product. However, it is important to note that the introduction of $Fe^{3+}$ promotes the photodegradation of NPM to produce more HONO and NOx. In the section below we suggest a tentative reaction mechanism to describe the formation of HONO and $NO_2$ upon irradiation of NPM at the water surface in the presence of soluble iron.

### 3.5 Mechanism Describing the Formation of HONO and NOx

We speculate that in the presence of $Fe^{3+}$, the decrease in dissolved nitrogen species that resulted from the photodegradation of NPM is the reason for the formation of HONO and NOx. Therefore, ROS and dissolved nitrogen containing ions were measured upon photodegradation of NPM in the presence of $Fe^{3+}$. The generation of superoxide radicals ($O_2\text{-}\bullet$), singlet oxygen ($^1O_2$) and hydroxyl radicals (OH) were quantified using DMPO, TEMP and DMPO as chemical probe molecules,

respectively. Figure 4A shows that in the absence of $Fe^{3+}$, the photodegradation of NPM induces generation of OH, $O_2$-•, and $^1O_2$, which can be ascribed to the electron transfer between the excited triplet state of NPM and the molecular oxygen ($O_2$) (Segura et al., 2008; Mostafa et al., 2013; Marin et al., 2012; Wang et al., 2021).

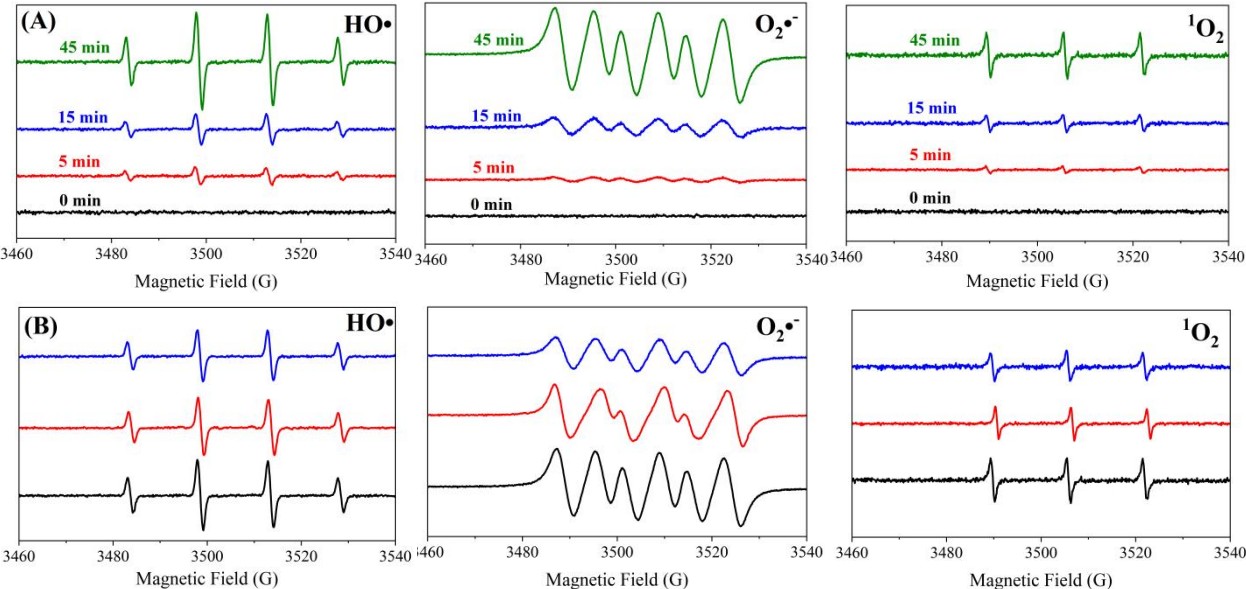

Figure 4. A) EPR spectra obtained upon photodegradation of NPM (0.5 mg $ml^{-1}$) in the absence of $Fe^{3+}$ as a function of the reaction time. B) EPR spectra obtained upon 45 min photodegradation of NPM (0.5 mg $ml^{-1}$) in the absence of $Fe^{3+}$ (dark line), and in the presence of 0.25 mg $ml^{-1}$ of $Fe^{3+}$ (red line) and 0.8 mg $ml^{-1}$ of $Fe^{3+}$ (blue line).

It has been reported that under UV light irradiation, $Fe^{3+}$ photo-reduction regenerates $Fe^{2+}$ accelerating the process due to the formation of new OH radicals (Segura et al., 2008). The EPR measurements revealed an interesting phenomenon that the increase of $Fe^{3+}$ concentration promotes the consumption rate of ROS (Figure 4B) rather than the production rate. The generated ROS would react with lower valence nitrogen-containing species to form HONO and NOx. Based on this finding we suggest a tentative reaction mechanism which could explain the formation of large quantities of HONO and NOx during the photochemical degradation of NPM. The photochemical generation of ROS could be driven by two pathways, pathway I: the excited tripled state of NPM ($^3NPM*$) can be formed under light irradiation (R1) (Mora et al., 2021), and then by reacting with water molecules (R2) it can trigger the formation of ROS such as OH radicals, accompanied by the generation of $O_2$-• through the transformation between radical anion of NPM (NPM-•) and dissolved oxygen (R3) (Wang et al., 2021). Furthermore, with the progress of the photodegradation of NPM, an increase of $O_2$-• and OH formation was observed (Figure 4A), favoring the HONO and $NO_2$ formation (R6-R8). In the presence of $Fe^{3+}$, the formation of OH radicals occur as well by R4 (Mazellier et al., 1997). In addition, nitrate ions ($NO_3^-$) and nitrite ions ($NO_2^-$) in the aqueous phase are formed by reactions R5 to R7. Peroxynitrate ($OONO_2^-$) is formed by reaction of $O_2$-• with $NO_2$, which thermally decomposes to form $NO_2^-$ and $O_2$ which further leads to HONO formation (R6) (Wang et al., 2020; Lammel et al., 1990; Goldstein et al., 1998). The reaction between $O_2$-• and NO can lead to the formation of $NO_2^-$ and $NO_3^-$, with a relatively fast rate constant of $4.3\times10^9$

M[-1] s[-1] (Goldstein et al., 1995) producing a peroxynitrite (OONO⁻) which then yields $NO_3^-$ through internal rearrangement (R7) (Løgager et al., 1993). At neutral pH (pKa=6.5), the product OONOH can also be formed by protonation, which can coexist with OONO⁻ to form $NO_2^-$ (R7) (Guptaet al., 2009). Previous studies have shown that the reaction between OH and $NO_2^-$ will generate $NO_2$ (R8) (Løgager et al., 1993), and sharp increase of HONO concentration occurs immediately from reaction between $NO_2^-$ and $H^+$ (R9), which is expected to be an important pathway of HONO formation.

At low $Fe^{3+}$concentrations (0.25-0.5 mg mL[-1]), the degradation rate of NPM was completely inhibited which was not the case for higher $Fe^{3+}$ concentration (0.5-0.8 mg mL[-1]) (Figure 1). Notably, $Fe^{3+}$ plays an important role in providing an acidic environment (pH=2.4-3.4) in the reaction system, which is followed by the redox reaction between $Fe^{2+}$ and $NO_3^-$ to produce $NO_2$ and consequently increase the amount of $NO_2$ (R10) (Figure S3). It has been shown that $NO_3^-$ undergo photochemical process and thus produces HONO (R11) and $NO_2$ (R12) (Ye et al., 2016; Zhou et al.,2011).

$$NPM \xrightarrow{h\nu} {}^3NPM^*  \qquad\qquad R(1)$$

$$^3NPM^* + H_2O \rightarrow H^+ + \cdot OH + NPM^{-\cdot} \qquad\qquad R(2)$$

$$NPM^{-\cdot} + O_2 \rightarrow NPM + O_2^{-\cdot} \qquad\qquad R(3)$$

$$Fe^{3+} + H_2O \xrightarrow{h\nu} Fe^{2+} + \cdot OH + H^+ \qquad\qquad R(4)$$

$$2NO_2 + H_2O \rightarrow HNO_2 + HNO_3 \qquad\qquad R(5)$$

$$O_2^{-\cdot} + NO_2 \rightarrow OONO_2^- \rightarrow O_2 + NO_2^- \xrightarrow{+H^+ \ (pKa=3.2)} HONO \qquad\qquad R(6)$$

$$O_2^{-\cdot} + NO \rightarrow OONO^- (NO_3^-) \leftrightarrows OONOH \xrightarrow{+OONO^-} O_2 + 2NO_2^- + H^+ \xrightarrow{pKa=3.2} HONO \qquad\qquad R(7)$$

$$NO_2^- + \cdot OH \rightarrow NO_2 + OH^- \qquad\qquad R(8)$$

$$NO_2^- + H^+ \xrightarrow{pKa=3.2} HONO \qquad\qquad R(9)$$

$$Fe^{2+} + NO_3^- + 2H^+ \rightarrow Fe^{3+} + NO_2(g) + H_2O \qquad\qquad R(10)$$

$$NO_3^- \xrightarrow{h\nu} [NO_3^-]^* \rightarrow O(^3P) + NO_2^- \xrightarrow{H^+} HONO \qquad\qquad R(11)$$

$$NO_3^- \xrightarrow{h\nu} [NO_3^-]^* \xrightarrow{H^+} NO_2 + \cdot OH \qquad\qquad R(12)$$

A simplified illustration of the reaction mechanism is shown in Figure S4. As shown in Figure S3, the HONO and $NO_2$ production during the photodegradation of NPM in the presence of $Fe^{3+}$ is significantly enhanced relative to that in the absence of iron ions. High $Fe^{3+}$ concentration (0.5-0.8 mg mL[-1]) promotes the HONO and $NO_2$ formation compared to low $Fe^{3+}$ concentrations (0.25-0.5 mg mL[-1]). The formed $NO_3^-$ and $NO_2^-$ were also measured by the ion chromatography analysis to evaluate the effect of $Fe^{3+}$ (see the details in Text S1 and Figure S5). As shown in Figure S5, the concentration of $NO_3^-$

and $NO_2^-$ decreased sharply in the presence of $Fe^{3+}$ compared to that in absence of $Fe^{3+}$. These results suggest that HONO and $NO_2$ enhancement during the irradiation of NPM solutions containing $Fe^{3+}$ can be ascribed to the transformation in the products distribution from $NO_3^-$ and $NO_2^-$ rather than a change in the products formation from the photodegradation of NPM.

## 4 Conclusions and outlook

Laboratory study revealed the formation of a greenhouse gas $N_2O$ by photolysis of NPM (Aregahegn et al., 2018), but previously the theoretical calculation predicted that the photolysis of NNs can generate $NO_2$ (Palma et al., 2020). The current study reveals that the light-induced degradation of NPM leads to enhanced production of HONO and NOx driven by secondary photochemistry between redox reaction of $Fe^{3+}/Fe^{2+}$ and photoproduced ROS. We quantified the photochemical HONO and NOx formation through NPM photodegradation, and we suggest that this chemistry may represent a significant source of HONO and NOx in the regions where surface waters are polluted with NNs insecticides. In order to estimate the relative importance of the NPM photolysis to global HONO and NOx emissions in the atmosphere, we parametrized the global HONO and NOx production related to NPM photochemistry, based on the NPM photolysis kinetic data and gridded downward solar radiation. The parameterization of HONO and NOx productions from NPM photolysis at $Fe^{3+}$ concentration of 0.025 mg $L^{-1}$ used in our estimation is based on Eq-S1, Eq-S2 and Eq-S3. The concentrations of NNs vary from several ng $L^{-1}$ to hundreds of µg $L^{-1}$ (Anderson et al., 2013). In view of the high concentration of NPM (50000 µg $L^{-1}$) used in our experiments, we selected a rationalization parameter scheme related to the environmental concentration of NPM (50 µg $L^{-1}$). The kinetic data has shown that the rate constant (k) is faster at low NPM concentration compared to that of high NPM concentrations (Figure S6). Current chemical models do not explicitly consider this source of reactive nitrogen species. In this manner, we are able to generate an hourly dataset of the NOx and HONO fluxes released from NPM chemistry, and we analyze the amount and spatial pattern of the fluxes in Figure 5. We note that although such estimation is rather simplified and can be biased in terms of the spatial heterogeneity as we do not consider the spatial variation of environmental NPM concentrations, our study presents a pioneer attempt to quantify the global source of HONO and NOx from the NPM photochemistry, as current chemical models do not explicitly consider this source of reactive nitrogen species. This inventory can be then applied in chemical models to quantify the environmental impact of HONO and NOx fluxes emerging from NPM photochemistry. The details about parameterization of HONO and NOx production emerged from NPM photochemistry are described in the Text S2. Figure 5 shows the spatial distributions of HONO and NOx fluxes produced from NPM photochemistry in the tested year 2017. The results indicate that globally produced HONO and NOx fluxes based on NPM photochemistry are 0.77 and 0.5 Tg N $year^{-1}$, respectively, making a total of 1.27 Tg N $year^{-1}$.

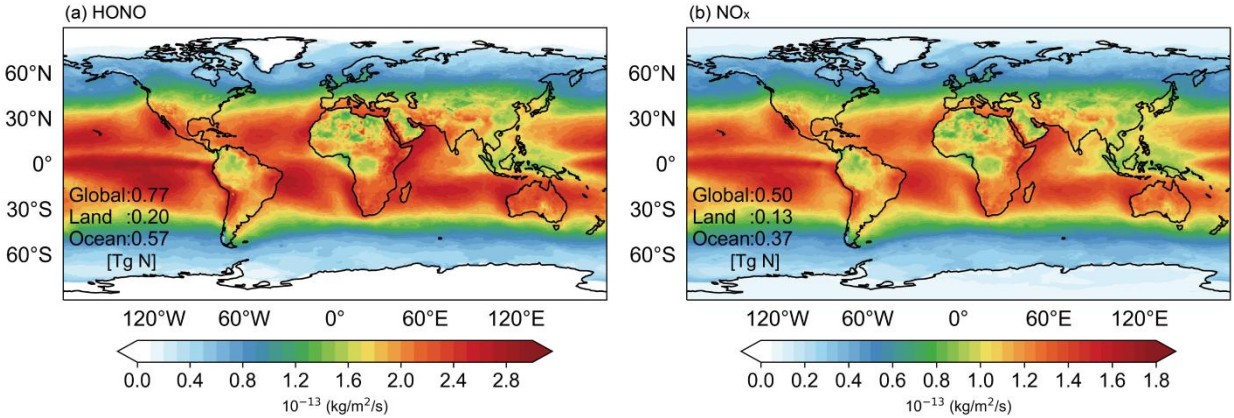

**Figure 5. Global emissions of HONO and NOx, produced by photochemistry of NPM in the presence of iron ions for year 2017.**

The total production of HONO and NOx emissions due to NPM photochemistry (1.27 Tg N year[-1]) represents 3.5% of the anthropogenic emissions of NOx related to fossil fuel in the year 2017 (36.2 Tg N year[-1], from the Community Emissions Data System (CEDS) inventory), and about 14.8% of the soil emissions (8.6 Tg N/year (Lu et al., 2021)). The highest HONO and NOx fluxes (74%) are produced by the photochemistry of NPM at the Ocean surface in the presence of iron ions, especially tropical oceans. The latter can be ascribed to the higher solar radiation in the tropic region. As displayed in the Figure S7, it is obvious to see that the spatial distribution of solar radiation is particularly strong in tropical oceanic regions, which can further confirmed the higher HONO and NOx fluxes at the ocean surface. The high reactive nitrogen emissions could also appear over other water surfaces like inland waters and lakes worldwide, through similar mechanisms induced by NPM photochemistry. Further studies are needed to quantify the relative importance of the recognized HONO and NOx sources from NPM photochemistry on a global scale as well as the impact on tropospheric ozone and OH in the marine boundary layer.

## Data availability

All raw data can be provided by the corresponding authors upon request.

## Author contributions

J.L. and S.G designed the research. Z.R., Y.L. and Y.H. performed the laboratory experiments. J.L., Z.R., X.G., S.L. C.Y., X.L. and S.G. analyzed and interpreted the data from laboratory experiments. Y.H., and Y.L. contributed to the relevant discussion on the manuscript. J.L., Y.H. and S.G. wrote the paper. All authors discussed the results and commented on the manuscript.

**Competing interests**

The authors declare no competing financial interest.

**Acknowledgments**

This work was supported by National Natural Science Foundation of China (No. 42207127, 42030712) and Applied Basic Research Foundation of Yunnan Province (Grant No. 202301AT070424, 202101BE070001-027, 202101BG070084). We are grateful to the National Natural Science Foundation of China (42177087 and 41977187), National Natural Science Foundation of China, Research Fund for International Scientists (4221101064), Yunnan Major Scientific and Technological Projects (Grant No. 202302AG050002), Yunnan Revitalization Talents Support Plan Young Talent Project and High-End Foreign Experts Project for financing this research.

**Supporting Information**

Additional 10 Figures, 3 Table and Supplementary Text. The supporting information is available free of charge via the Internet on the ACS Publications website at http://pubs.acs.org.

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
