# Peer review of "Formation of Reactive Nitrogen Species Promoted by Iron Ions Through the Photochemistry of Neonicotinoid Insecticide"

_EGUsphere, 2024_

## Author Comment (AC1)

We appreciate the careful consideration and constructive comments of the reviewer on this manuscript. We have carefully responded to all **point-by-point comments** and issues and have revised the manuscript accordingly. The revisions are described in details below.

**Response to Reviewer #1:**

The study by Ran et al. outlines laboratory experiments aimed at studying the gaseous products emitted when the neonicotinoid pesticide nitenpyram (NPM) is photolyzed in aqueous solutions in the presence of iron(III). Analysis of the gas phase species NO and $NO_2$ were performed using a chemiluminescence detector with a molybdenum converter, while the carbonate denuder method was used to measure HONO. Aqueous phase reactive oxygen species such as superoxide and hydroxyl radical were found to be present using EPR. The main finding of this study is that in the presence of iron, organic nitrogen functional groups found in NPM are converted to inorganic species such as NOx and HONO, which outgas from solution and are detected in the gas phase. Some kinetic studies were carried out, which in principle can be used to assess the environmental (photochemical) fate of this pesticide. The authors also carried out a modeling exercise to determine the impact of NPM on global atmospheric HONO and NOx budgets. The novelty of the work is not so clear, although there appears to be some previous work done to show that neonicotinoids photolyze more rapidly in the presence of iron. Neonicotinoid pesticides are a major environmental hazard due to the massive quanitities they are applied to ag fields and their toxicity to pollinators. So, I would say understanding the fate of neonics in general is more important than its impact on HONO and NOx, which is likely quite minor. The experimental methods are somewhat standard, although it appears that little attention was paid to controlling pH of the solution.   The most significant problem in the paper was how the modeling work done, the lack of detail provided, and what seem to be nonsensical results. In my opinion, the study requires significant work yet, especially on the modeling side.

**Response:** Thanks for the constructive comments and good suggestions.

**Specific Comments**

Introduction: There needs to be a more thorough discussion of the previous work done on neonicotinoid pesticide photochemistry in the introduction section. There are a number of articles now on the topic, including showing how $Fe^{3+}$ can catalyzed the photodegradation (some of these are cited later in the paper).

**Response:** As suggested, we have added the discussion of the previous work done on neonicotinoid pesticide photochemistry. In the revised manuscript (in line 57), we added the the following paragraph: "It has been shown that the ozonolysis of NNs insecticides on various surfaces could contribute to the formation of gaseous nitrous acid (HONO) (Wang et al., 2020). Gaseous nitrous oxide ($N_2O$), which is a potent greenhouse gas, was previously identified as the gas-phase product in the photolysis of solid thin films of NNs (Nitenpyram, acetamiprid, thiamethoxam, thiacloprid,

clothianidin and dinotefuran), with yields of $\Delta N_2O/\Delta NN > 0.5$ in air at both 313 and 254 nm (Wang et al., 2019; Aregahegn et al., 2017; Aregahegn et al., 2018). Palma et al (Palma et al., 2020) used a gas-flow reactor connecting with a NOx analyzer, and the production of gaseous $NO/NO_2$ was founded during irradiation (300-450 nm) of imidacloprid."

In line 83, we have also added the discussion of the previous work done on photodegradation of NNs in the presence of $Fe^{3+}$. In the revised manuscript, we added the following paragraph "Previous studies focused on NNs oxidation in the ferric aqueous phase have focused on the photo-Fenton reaction (Malato et al., 2021; Lacson et al., 2018; Wang et al., 2022; Nguyen et al., 2020; Sedaghat et al., 2016) and heterogeneous-phase photocatalysis (Rózsa et al., 2019; Sun et al., 2019; Hayat., 2019; Soltani-Nezhad., 2019). As recently reported, the photolysis of iron can generate several reactive oxygen species (ROS) such as $O_2^{\cdot-}/HO_2^{\cdot}$, which can trigger the redox cycle between $Fe^{3+}$ and $Fe^{2+}$ (Gen et al., 2021) as well as promote NNs insecticides oxidation. Meanwhile, iron ions inhibit the degradation of organic matter through the formation of complexes, mainly due to fluorescence bursting. The complexation may cause the inhibition of the excited singlet state and thus the photoformation of the excited tripled state (Wan et al., 2019). "

Abstract: In justifying the importance of this paper in the abstract, the authors claim that neonicotinoid pesticides are a source of HONO and NOx during the daytime that can help "narrow the gap between field observations and model outcomes of HONO in the atmosphere…can be an important contribution to the global nitrogen cycle affecting the atmospheric oxidizing capacity as well as climate change." As noted below, I believe the stated importance of neonicotinoid pesticides as a source of HONO and NOx may need to be reevaluated. Despite this, I think any research on the fate of neonicotinoid pesticides is useful in that these compounds are used extensively on crops and due to their deleterious impact on pollinator populations. Thus, I think it may be wiser to redirect the focus of their paper to the fate of the neonicotinoid pesticides in general. In this context, HONO and $NO_x$ are simply photodegradation products and the experiments discussed are used to deduce the degradation mechanism. The quantum yields and photolysis rates calculated under actinic conditions can then be used to simply calculate a photolysis half-life. This would be much more useful than an erroneous global model where proper model inputs are not available or have huge errors associated with assumptions made.

**Response:** We are thankful to the reviewer for this suggestion but we still would like to keep the focus on HONO and NOx produced by the photodegradation of Nitenpyram. This study represents as a pioneer attempt to quantify the global source of HONO and NOx from the NPM photochemistry, as current chemical models do not explicitly consider this source of reactive nitrogen species.

The scientific arguments for this statement are described below.

Photolysis is the main natural degradation pathway for NNs, and their photolytic organic products and environmental fate have been explored by many researchers (Lu et al., 2015; González-Mariño et al., 2018; Aregahegn et al., 2017; Aregahegn et al., 2018). However, there is a lack of specific quantification of the reactive nitrogen gas species produced by NNs photolysis. Currently, there are still unknown sources of atmospheric reactive nitrogen species (HONO and NOx) and the mechanisms of their generation have not been fully clarified, and in particular the much higher levels of HONO observed during daytime than predicted by the model are still not fully explained (Jiang et al., 2020; Lee et al., 2016; Wang et al., 2021; Song et al., 2023). It has been reported that photolysis of nitro compounds is an important source of HONO (Yang et al., 2021; Bejan et al., 2006; Fukuhara et al., 2001). Yang et al (Yang et al., 2021) has reported that the photolysis frequency for HONO varied in the range of $10^{-7}$ $s^{-1}$ and $10^{-5}$ $s^{-1}$ by nitrophenols and nitro polycyclic aromatic hydrocarbons, respectively. NPM, as a typical NNs, belongs to a nitro pesticide [(R1N)(R2N)C=CHNO2] compound. Little is known about the accurate assessment of the gas phase products and corresponding reaction mechanism.

Herein, we focused on quantifying the photolytic production of reactive nitrogen species from NPM in the presence of soluble iron. As we demonstrated by our laboratory experiments the NPM photolysis releases large amounts of HONO and NOx in the presence of soluble iron, which provides an important source of HONO and indeed helps to narrow the gap between field observations of atmospheric HONO and model results.

Line 146:   The authors state, "The change of $Fe^{3+}$ concentrations may alter the pKa of the acid-base balance…" This doesn't seem to make chemical sense and needs revision/clarification. The pKa is a property of a weak acid that does not depend necessarily on the $Fe^{3+}$ concentration unless the iron salts are changing the ionic strength considerably. Rather it is the pH that may alter the speciation of the $Fe^{3+}$. Perhaps the authors meant that the addition or presence of $FeCl_3$ changes the pH of the system, which would affect the speciation of NPM?

**Response:** We apologize for the misunderstanding. In this study, we found that the introduction of iron provided a strongly acidic environment for the system by testing the pH of the solution. Also in some previous studies, it has been reported (Zhou et al., 2019; Cai et al., 2018) that a change in pH can have a slight effect on the absorption spectrum of a solution. We have revised the statement of "The change of $Fe^{3+}$ concentrations may alter the pKa of the acid-base balance…" to "The change of $Fe^{3+}$ concentrations may alter the pH of the system, which in turn may affect the protonation/deprotonation degree of NPM, and therefore affects its absorption spectrum (Zhou et al., 2019)."

 Line 150:   I recommend being more quantitative here. i.e., estimate the proportion of weak acid and conjugate base present.

**Response:** Thanks for your suggestion but unfortunately, the calculation of the proportion of weak acid and conjugate base present are unavailable at current conditions. However, it should be noted that with the introduction of iron, it can be provided a strongly acidic environment for the system, which in turn may affect the protonation/deprotonation degree of NPM, and therefore affects its absorption spectrum.

Section 3.2.   It is stated that iron is present in natural waters at concentrations ranging between $10^{-7}$ M and $1 \times 10^{-4}$ M. What kind of natural waters? Aerosols, marine, rivers? Please explain why such high concentrations of iron were used for experiments. It is especially important for connecting to the modeling study.

**Response:** According to the previous studies (Li et al., 2018), we can state that iron ions are ubiquitous in natural waters with concentrations ranging between $10^{-7}$ and $10^{-4}$ M. The concentrations of Fe in raindrops, fog and cloud waters range from 0.1 to $1138 \times 10^{-6}$ M, depending on the location with typical concentration in cloud droplets of about $10^{-6}$ M (Deguillaume et al., 2005). Below we summarized the distribution of soluble iron concentrations in the waters as shown in Table R1 and also Table S3 in the supplementary information. The higher iron concentration selected in this paper is mainly to reduce the phenomenal error caused by too low iron concentration. At the same time, at a relatively high iron concentration level, we can build a reasonable relationship between HONO and NOx formation and different NPM concentrations, which is helpful to establish parameterization schemes. However, in order to estimate the environmental NPM and iron concentration contributed to the formation of reactive nitrogen species, we selected a rationalization parameter scheme related to the environmental concentration of NPM (50 µg L$^{-1}$) and soluble iron (92.48 nmol L$^{-1}$, 0.025 mg L$^{-1}$ in our study), which is representative of certain significance.

In the revised SI, we have added Table R1 as Table S3 and updated the sentence "We summarized the distribution of soluble iron concentrations in the waters as shown in Table S3 in the supplementary information. In order to estimate the environmental NPM and iron concentration contributed to the formation of reactive nitrogen species, we selected a rationalization parameter scheme related to the environmental concentration of NPM (50 µg L$^{-1}$) and soluble iron (92.48 nmol L$^{-1}$, 0.025 mg L$^{-1}$ in our study), which is representative to a certain significance."

**Table R1.** The concentrations of soluble iron in water of different region

| Sampling site | Concentration (nmol L$^{-1}$) | References |
|---|---|---|
| Humic acid-rich coast of the Pacific Ocean | 23.1-573.2 | Gerringa et al., 2007 |
| Scheldt estuary | 104-536 | Batchelli et al., 2010 |
| Peconic Estuary | 9-240 | Gobler et al., 2002 |

| | | |
|---|---|---|
| Southern Vancouver Island | 0.05 – 0.07 | Nishioka et al., 2001 |
| Alaskan coastal waters | 0.5-4.1 | Lippiatt et al., 2010 |
| the outer bay of Mediterranean coastal waters | 3.7-25 | |
| in the inner and middle bay of Mediterranean coastal waters | 9-240 | Öztürk et al., 2003 |
| Yamuna River in Mathura | 1.73 mg L$^{-1}$ | Ahmed et al., 2022 |

Line 198: The authors mention they quantified quantum yields of HONO, NO$_2$, and NO formation from NPM photolysis from the measured photolysis frequencies (J-values). However, they never report what they are, nor do they discuss them in the context of prior studies for related chemicals to evaluate whether they make sense or not.

**Response:** The photolysis frequency (J-values) had been used by Yang et al (Yang et al., 2021) to evaluate the quantification of nitroaromatic compounds photolysis to form HONO and NOx. In order clarify the photolysis frequency in the revised manuscript, we added some discussions in Line 226 as following: Previous studies (Yang et al., 2021) have found that the photolysis frequency of HONO and NO in nitrophenol solid-phase films (4-nitrophenol, 4-nitrocatechol, 3,5-dinitrosalicylic acid, 3-nitrosalicylic acid, and 5-nitrosalicylic acid) varied in the ranges of (0.34-4.16)×10$^{-7}$ and (0.38-3.21)×10$^{-7}$ s$^{-1}$, respectively, when irradiated by Xenon lamps. NPM liquid-phase photolysis produced HONO and NOx at the photolysis frequency of 10$^{-7}$, but the addition of iron resulted in the photolysis frequency of 10$^{-6}$ for HONO, suggesting that iron significantly facilitated the release of HONO.

Line 278: The statement, "but the theoretical calculation predicted that the photolysis of NPM can generate NO$_2$ rather than N$_2$O. What is this referring to? This seems to refer to a theory study. If this refers to the Aregahegn study, then clarify that this work also included calculations so it is clear.

**Response:** We apologize for the misunderstanding. We have modified the sentence as follows: "Laboratory study revealed the formation of a greenhouse gas N$_2$O by photolysis of NPM (Aregahegn et al., 2018), but previously the theoretical calculation predicted that the photolysis of NNs can generate NO$_2$ (Palma et al., 2020)."

Figure S4 suggests that nitrate is the major product in the absence of Fe$^{3+}$. This is not explained clearly. Where does the nitrate come from?

**Response:** The formation of nitrate and nitrite in the absence of $Fe^{3+}$ was described in the sentences starting from line 37 in the supplementary text. "$NO_2$ produced by direct photolysis of NPM is hydrolyzed in aqueous media to form $NO_2^-$ and $NO_3^-$, and nitrogenous species are partially dissolved during their release from the liquid to the gas phase. The reaction of $NO_x$ with $O_2^{-\cdot}$ radicals will also produce $NO_2^-$ and $NO_3^-$".
Also the R5-R7 describing in the manuscript have indicated the formation of nitrate and nitrite.

$$2NO_2 + H_2O \rightarrow HNO_2 + HNO_3 \qquad\qquad R(5)$$

$$O_2^{-\cdot} + NO_2 \rightarrow OONO_2^- \rightarrow O_2 + NO_2^- \xrightarrow{+H^+ \ (pKa=3.2)} HONO \qquad\qquad R(6)$$

$$O_2^{-\cdot} + NO \rightarrow OONO^- \ (NO_3^-) \leftrightharpoons OONOH \xrightarrow{+OONO^-} O_2 + 2NO_2^- + H^+ \xrightarrow{pKa=3.2} HONO \qquad R(7)$$

Page 11: the authors attempt to evaluate the importance of NPM as a source of HONO and NOx in the atmosphere by using what appears to be a global model. This is the most problematic part of the paper due to the fact that the model is so poorly described. There seems to be little effort to thoroughly document the experimental and modeling methods and procedures to allow their work to be reproduced at a later date. For example, there are a total of 10 lines of text in the supplemental information file describing this model with very little detail that anyone can evaluate the validity of the approach. Moreover, there is no literature cited-nothing to aid in reproducing their modeling results besides the equations used to parameterize HONO and NOx emission rates in the model. Also, the equations are written in an ambiguous manner with no attention to defining terms or units. It is not possible to evaluate whether the model was run appropriately. Any parameterization of photochemistry would need to account for wavelength-dependent molar absorption coefficients, quantum yields, and actinic flux. While there is mention of how the model models actinic flux, there is no information on which absorption coefficients or quantum yield(s) are used (nothing is tabulated for someone who would like to reproduce the experiments). Are there any assumptions being made in their calculation and if so, what are they? How does the model account for fluctuations in iron content on a global scale? What types of surfaces in the model are the sites for these reactions? Aerosols, soils, foliage, buildings? How are those surfaces represented? Such a model needs to account for mass loading of the pesticide on fields. This requires geospatial data of pesticide usage on a global scale. Yet, there is not discussion of the data set that was used for this. Also, once applied to the fields, there will be a fraction of the pesticide that is absorbed by plants and soil and not available for photolysis. The way it is written, it seems the authors treat the entire globe as an aqueous reactor where the aqueous concentration of NPM is a uniform 50 microgram per liter.

**Response**: We apologize for the confusion. Here, we do not apply a "model" which typically refers to an atmospheric chemical model in which the emissions, transport, chemistry, and deposition of NOx and HONO generated from NPM can be quantified.

Instead, we are estimating the amount of NOx and HONO fluxes released from NPM chemistry, as a function of NPM concentration and solar radiation, following Eq-S1, Eq-S2 and Eq-S3, but assuming that the environmental NPM concentration is three orders smaller (50 µg L$^{-1}$) than the experimental conditions of 50000 µg L$^{-1}$. The parameterization of HONO and NOx productions from NPM photolysis at $Fe^{3+}$ concentration of 0.025 mg L$^{-1}$ used in our estimation is based on Eq-S1, Eq-S2 and Eq-S3. The estimation is conducted for each of the 561×360 grids at the globe with a horizontal resolution of 0.5×0.625°, consistent with resolution of the solar radiation data from the hourly Modern-Era Retrospective analysis for Research and Applications Version 2 (MERRA2) reanalysis dataset.

$Y_{HONO}=1.58595*10^9X-1.19123*10^{11}$                              Eq-S1

$Y_{NO2}=6.58261*10^8X-1.81889*10^{10}$                               Eq-S2

$Y_{NO}=2.58054*10^8X-1.41507*10^{10}$                                Eq-S3

where Y( molecules cm$^{-2}$ s$^{-1}$) represents the HONO/NOx fluxes, X (W m$^{-2}$) represents the light density.

A key procedure is to consider the concentration of NPM at each of the 0.5×0.625° grids. Ideally, the NPM concentration should display spatial distributions as the iron contents, solar light intensity and underlying surface is different for each region. However, As we all known, neonicotinoid pesticides (NNs) account for a large fraction of the total world insecticide market and they are widely distributed throughout the environment including various surfaces, water, vegetation, soil and dust particles. Many studies are available in the literature focused on the reactions and fate of NNs in aqueous systems (Bonmatin et al., 2014; Morrissey et al., 2015; Borsuah et al., 2020), and recent studies have reported that the concentrations of neonicotinoids are higher than historically used organophosphate insecticides in the aqueous phase (Hladik et al., 2014), and the concentrations of some neonicotinoids reached 44.1 µg L$^{-1}$ in lake water, Texas, USA (Anderson et al., 2013). Although the photolysis products and mechanism of neonicotinoids have been investigated using either simulated sunlight irradiation under laboratory conditions or by use of natural sunlight (Lu et al., 2015; Aregahegn et al., 2017), yet little is known about the accurate assessment of the gas phase products and corresponding reaction mechanism, especially the reactive nitrogen species. Based on the above information, our study reveals that the light-induced degradation of NPM leads to enhanced production of HONO and NOx driven by secondary photochemistry between redox reaction of $Fe^{3+}/Fe^{2+}$ and photoproduced ROS. We quantified the photochemical HONO and NOx formation through NPM photodegradation, and we suggest that this chemistry may represent a significant source of HONO and NOx in the regions where surface waters are polluted with NNs insecticides. In addition, according to one review study, some studies related to rivers pollution in Canada, Europe China, and the U.S. observed concentration levels of NPM above the chronic concentration limits for aquatic systems (Figure R1-cite from Borsuah's study), and there are 25 studies that

reported the concentration of NNs in the range from 38.2 to 190.4 µg L$^{-1}$. We have to stress that our experiment is not able to derive the relationship between HONO, NOx emissions and light density at NPM concentration lower than 50000 µg L$^{-1}$ due to the current limit of detection. As such, we assume that the environmental NPM concentration is three orders smaller (50 µg L$^{-1}$) than the experimental conditions of 50000 µg L$^{-1}$, and do not consider its spatial heteorogenity in the model. As a result, the variation of NOx and HONO emissions is driven by the solar radiation.

Furthermore, the widespread use of NPM and its capability to release HONO and NOx suggests that NPM might be an unexplored source of global atmospheric reactive nitrogen (Nr) and hence influence air quality and climate. Evaluation of such impacts requires a parameterization of global HONO and NOx fluxes emerging from NPM photochemistry in chemical transport models. However, current chemical models do not explicitly consider this source of reactive nitrogen species. In this manner, we are able to generate an hourly dataset of the NO$_x$ and HONO fluxes released from NPM chemistry, and we analyze the amount and spatial pattern of the fluxes in Figure 5. We note that although such estimation is rather simplified and can be biased in terms of the spatial heterogeneity as we do not consider the spatial variation of environmental NPM concentrations, our study presents a pioneer attempt to quantify the global source of HONO and NOx from the NPM photochemistry, as current chemical models do not explicitly consider this source of reactive nitrogen species. This inventory can be then applied in chemical models to quantify the environmental impact of HONO and NOx fluxes emerging from NPM photochemistry.

In terms of the reproductivity, we have uploaded Python codes for generating the global flux for your review and should be able to be public upon the acceptance of this manuscript. The code can be found in github.

[Figure]

**Figure R1.** Neonicotinoid pesticides concentrations detected in global aquatic systems (adapted from Borsuah's study).

We have now added the statement in the revised manuscript to provide more information of the flux estimation in Text S2: We are estimating the amount of NOx and HONO fluxes released from NPM chemistry, as a function of NPM concentration and solar radiation, following Eq-S1, Eq-S2 and Eq-S3, but assuming that the environmental NPM concentration is three orders smaller (50 µg L$^{-1}$) than the experimental conditions of 50000 µg L$^{-1}$. The parameterization of HONO and NOx productions from NPM photolysis at Fe$^{3+}$ concentration of 0.025 mg L$^{-1}$ used in our estimation is based on Eq-S1, Eq-S2 and Eq-S3 (Figure S8 and Figure S9). The estimation is conducted for each of the 561×360 grids at the globe with a horizontal resolution of 0.5×0.625°, consistent with resolution of the solar radiation data from the hourly Modern-Era Retrospective analysis for Research and Applications Version 2 (MERRA2) reanalysis dataset.

$Y_{HONO}=1.58595*10^9X-1.19123*10^{11}$               Eq-S1

$Y_{NO2}=6.58261*10^8X-1.81889*10^{10}$                Eq-S2

$Y_{NO}=2.58054*10^8X-1.41507*10^{10}$                Eq-S3

Where Y( molecules cm$^{-2}$ s$^{-1}$) represents the HONO/NOx fluxes, X (W m$^{-2}$) represents the light density.

A key procedure is to consider the concentration of NPM at each of the 0.5×0.625° grids. Ideally, the NPM concentration should display spatial distributions as the iron contents, solar light intensity and underlying surface is different for each region.

We have to stress that our experiment is not able to derive the relationship between HONO, NOx emissions and light density at NPM concentration lower than 50000 µg L$^{-1}$ due to the current limit of detection. As such, we assume that the environmental NPM concentration is three orders smaller (50 µg L$^{-1}$) than the experimental conditions of 50000 µg L$^{-1}$, and do not consider its spatial heterorogenity in the model. As a result, the variation of NOx and HONO emissions is driven by the solar radiation. Furthermore, the widespread use of NPM and its capability to release HONO and NOx suggests that NPM might be an unexplored source of global atmospheric reactive nitrogen (Nr) and hence influence air quality and climate. Evaluation of such impacts requires a parameterization of global HONO and NOx fluxes emerging from NPM photochemistry in chemical transport models. However, current chemical models do not explicitly consider this source of reactive nitrogen species. In this manner, we are able to generate an hourly dataset of the NOx and HONO fluxes released from NPM chemistry, and we analyze the amount and spatial pattern of the fluxes in Figure 5. We note that although such estimation is rather simplified and can be biased in terms of the spatial heterogeneity as we do not consider the spatial variation of environmental NPM concentrations, our study presents a pioneer attempt to quantify the global source of HONO and NOx from the NPM photochemistry, as current chemical models do not explicitly consider this source of reactive nitrogen species. This inventory can be then applied in chemical models to quantify the environmental impact of HONO and NOx fluxes emerging from NPM photochemistry.

In lines 152-160 in the revised manuscript, we have also add the sentence:

2.7 Global simulation of NOx and HONO fluxes

We estimated the global inventory of the NOx and HONO fluxes produced by NPM photochemistry using the observation-constrained parametrization scheme and hourly solar radiation data. Gridded and hourly downward solar radiation data are obtained from the Modern-Era Retrospective analysis for Research and Application version 2 (MERRA-2) assimilated meteorological fields. We calculated the flux of NOx and HONO for each model grid at a horizontal resolution of $0.5° \times 0.625°$ (consistent with MERRA2 radiation dataset) following Eq-S1, Eq-S2 and Eq-S3, but assuming that the environmental NPM concentration is three orders smaller than the experimental conditions of 0.5 mg $L^{-1}$. The parameterization of HONO and NOx productions from NPM photolysis at $Fe^{3+}$ concentration of 0.025 mg $L^{-1}$ used in our estimation is based on Eq-S1, Eq-S2 and Eq-S3.

In lines 320-327 in the revised manuscript, we have also add the sentence:

Current chemical models do not explicitly consider this source of reactive nitrogen species. In this manner, we are able to generate an hourly dataset of the NOx and HONO fluxes released from NPM chemistry, and we analyze the amount and spatial pattern of the fluxes in Figure 5. We note that although such estimation is rather simplified and can be biased in terms of the spatial heterogeneity as we do not consider the spatial variation of environmental NPM concentrations, our study presents a pioneer attempt to quantify the global source of HONO and NOx from the NPM photochemistry, as current chemical models do not explicitly consider this source of reactive nitrogen species. This inventory can be then applied in chemical models to quantify the environmental impact of HONO and NOx fluxes emerging from NPM photochemistry.

Figure 5: Geospatial emissions of HONO and NOx are presented showing high emissions mostly over the oceans and in the mid latitudes. Numbers are written on the map as, "Global: 0.77; Land: 0.20; and Ocean 0.57). First of all, it is not clear what those numbers refer to as there are no units associated with them. I am assuming they are in Tg N/y since the values somewhat resemble those stated in the text. These numbers are nonsensical for the fact that neonicotinoids are not applied to the World's oceans. They are applied to cropland and mostly in the northern hemisphere. There is enough resolution on the map to allow one to identify the World's major agricultural regions and it is puzzling why they show less emissions of HONO and NOx due to NPM photochemistry than do the oceans. This indicates a major flaw in the model. For this reason, I do not trust the statement on line 294, which states that NPM photochemistry represents 3.5% of the total anthropogenic emissions of NOx related to fossil fuel in the year 2017.

**Response:** In the Figure 5, numbers are written on the map as, "Global: 0.77; Land: 0.20; and Ocean 0.57". The results indicated that globally produced HONO and NOx fluxes based on NPM photochemistry are 0.77 and 0.5 Tg N $year^{-1}$, respectively, making a total of 1.27 Tg N $year^{-1}$. We have add the units associated with them in the revised Figure 5 (see below).

[Figure]

**Figure R2.** Global emissions of HONO and NOx, produced by photochemistry of NPM in the presence of iron ions for year 2017.

As previous studies reported, the neonicotinoids are widely applied in cropland and soil, however, only about 5% of the applied neonicotinoids reach the target pest and over 90% of the applied active components of neonicotinoids end up in the soil. Neonicotinoids are highly water soluble in soil, making them susceptible to losses during agricultural runoff and drainage, then high concentrations of neonicotinoids have been found in various environmental conditions. In addition, according to one review study, some rivers studies completed in Canada, Europe China, and the U.S. observed concentration levels above the chronic concentration limits for aquatic systems (Figure above), and we can see there are 25 studies which observed the concentration of NNs ranging from 38.2 to 190.4 µg L$^{-1}$. Due to the great variation and complexity of the concentration distribution of neonicotinoid in the environment, in this study, we selected a more appropriate concentration value (50 µg L$^{-1}$) in the environment. In this manner, we are able to generate an hourly dataset of the NO$_x$ and HONO fluxes released from NPM chemistry, and we analyze the amount and spatial pattern of the fluxes in Figure 5. We note that although such estimation is rather simplified and can be biased in terms of the spatial heterogeneity as we do not consider the spatial variation of environmental NPM concentrations, our study presents as a pioneer attempt to quantify the global source of HONO and NOx from the NPM photochemistry, as current chemical models do not explicitly consider this source of reactive nitrogen species. This inventory can be then applied in chemical models to quantify the environmental impact of HONO and NOx fluxes emerging from NPM photochemistry.

The highest HONO and NOx fluxes (74%) are produced by the photochemistry of NPM at the ocean surface in the presence of iron ions, especially tropical oceans. The reason can be ascribed to the higher solar radiation in the tropic region, as displayed in the Figure R3 and Figure S7, it is obvious to see that the spatial distribution of solar radiation is particularly strong in tropical oceanic regions, which can further confirm the higher HONO and NOx fluxes at the ocean surface.

We have now added the statement in the revised manuscript to provide more information of the emissions of HONO and NOx in line 338-340: As displayed in the Figure S7, it is obvious to see that the spatial distribution of solar radiation is

particularly strong in tropical oceanic regions, which can further confirm the higher HONO and NOx fluxes at the ocean surface.

[Figure]

**Figure R3.** The spatial distribution of solar radiation in the global region.

This estimation gives an idea about the importance of the photochemical processes of NNs leading to HONO and $NO_2$ formation on a global level indicating that further laboratory studies should evaluate more in details this chemistry at different experimental conditions. The emerged outcomes then can be modelled on a regional and global scale to assess the influence of NNs photochemistry on the topical issues in 21st century such as air quality and climate change.

Experimental section: Experimental details are lacking in some areas. What is the volume of solution used? The pH? Was the reactor static or a flow reactor? Was the system stirred? Would be useful to provide a figure showing how the experiment was carried out. The authors appear to use a spectroradiometer to measure spectral irradiance. How was the irradiance measurement conducted? Does this account for light absorption through the walls/window of the reactor? Can the authors be sure that the light was not attenuated too much by the high concentrations of solutes in the reactor to the point that the reactor depth was not evenly irradiated? Why did the authors use a carbonate denuder to measure HONO? How did the authors account for breakthrough, which is a major problem with this method. Also, how did the authors correct for the HONO that is converted to NO on the Mo catalyst and would lead to an erroneous $NO_2$ signal?

**Response:** In our experimental system, solid NPM (Aladdin, China) was dissolved in ultra-pure water to prepare an aqueous NPM solution (0.5 mg mL$^{-1}$). FeCl$_3$ (98%, Aladdin China) was used as the source of different concentrations of aqueous Fe$^{3+}$ (0.1-0.8 mg mL$^{-1}$), and their solutions were prepared by dissolving the corresponding mass of FeCl$_3$ in ultra-pure water. A 50 mL solution was placed on the circular reactor

before each experiment. The pre-reaction initial pH of NPM was 7.3, and $Fe^{3+}$ plays an important role in providing an acidic environment (pH=2.4-3.4) in the pre-reaction system (Table S1). The reaction solution was static during the reaction without stirring the solution. We have added a diagram of the experimental setup in the part of supporting information as shown in Figure R4 (Figure S1 in the revised manuscript).

[Figure]

**Figure R4.** Diagram of the experimental set up.

The spectral irradiance of the xenon lamp was measured by a calibrated spectroradiometer (HP 350 UVP, China). As displayed in the Figure R5, the spectrogram from the direct measurement (black one) shows little difference compared to the one through the upper walls/window of the reactor (red one). However, we also conducted the measurements of spectrogram corresponded to spectral irradiance through the whole walls/window of the reactor (blue one), and the intensity of the spectral irradiance was slightly attenuated by the reactor.

[Figure]

**Figure R5.** The comparison of the spectral irradiance emitted by Xenon-lamp

NO, $NO_2$ and HONO concentrations were detected using a chemiluminescence NOx analyzer (42i, THERMO) with a molybdenum converter. Because HONO was detected by a quartz tube (25 cm length, 2.9 cm inner diameter) filled with $Na_2CO_3$ between the reactor and the analyzer was employed to remove HONO. The removal efficiency of HONO by the $Na_2CO_3$ tube reached 99% at the steady sate (Han et al.,

2016). This HONO detection technique has been widely employed to measure the HONO concentration (Han et al., 2016; Yang et al., 2020; Cazoir et al., 2014; Brigante et al., 2008; Monge et al., 2010; Zhou et al., 2018). Meanwhile, it should be noted that $NO_2$ and $NO$ are barely captured by the $Na_2CO_3$ tube as shown in the Figure R6. In order to validate the feasibility of the method, we also performed additional test experiment to confirm the indirectly determined HONO values by using Water-Based Long-Path Absorption Photometer (WLPAP, Beijing Zhichen Technology Co., Ltd, China) on-line connected with the reactor for real-time measurements of HONO, and the results agree well with the performed measurements by $Na_2CO_3$ tube (Figure R7).

[Figure]

**Figure R6.** The adsorption of $NO_2$ and $NO$ in the reactor by $Na_2CO_3$ tube.

[Figure]

**Figure R7.** Typical HONO profile measured in real time by WLPAP analyzer upon irradiation of NPM. Conditions: Irradiation intensity of 169.4 W $m^{-2}$ at 300< $\lambda$ <400 nm, NPM concentration of 0.1 mg $ml^{-1}$, temperature of 298 K.

Comment on title: The title should be revised to read as: "Formation of Reactive Nitrogen Species Promoted by Iron Ions through the Photochemistry of a Neonicotinoid Insecticide. Note the use of the indefinite article before Neonicotinoid since the authors only studied one insecticide, meaning it would not be correct to

assume that the chemistry applies to all neonicotinoid insecticides without additional evidence.

**Response:** We have revised the title to "Formation of Reactive Nitrogen Species Promoted by Iron Ions through the Photochemistry of a Neonicotinoid Insecticide".

Comment on short summary: The authors' use of 'eruptive' is not appropriate in this context. I suggest revising the sentence to: We report the enhanced formation of nitrous acid (HONO) and NOx (NO + $NO_2$) during the photolysis of a neonicanoid insecticide in the presence of iron at the air-water interface.

**Response:** We have revised the short summary as suggested "We report the enhanced formation of nitrous acid (HONO) and NOx (NO + $NO_2$) during the photolysis of a neonicotinoid insecticide in the presence of iron at the air-water interface".

Line 303: Regarding data availability, the authors should provide tables of wavelength-dependent absorption and quantum yields to allow readers to use this data in the future to either conduct their own research or to check over the published work.

**Response:** The quantum yield ($\phi$) is a characteristic parameter defining how efficiently a compound degrades upon absorption of a photon. As suggested, we provided the quantum yields of HONO and NOx produced by NPM photolysis between 300-400 nm as in Table R2, and the absorption cross sections of NPM at different iron concentrations as in Figure R8. Meanwhile, the calculation of the quantum yield is added in the text section of the supporting information. The added content is as follows:

Text S3. Calculation of Quantum Yields

The quantum yields for reactive nitrogen species formation ($\Phi$), including $\Phi_{HONO}$, $\Phi_{NOx}$ and $\Phi_{NPM}$ can be determined by the equation (Eq-S4 and Eq-S5).

$$\emptyset_{HONO} = \frac{J_{NPM \to HONO}}{\int_{\lambda_1}^{\lambda_2} I(\lambda)\,\sigma(\lambda)d\lambda} \qquad\qquad Eq\text{-}S4$$

$$\emptyset_{NOx} = \frac{J_{NPM \to NOx}}{\int_{\lambda_1}^{\lambda_2} I(\lambda)\,\sigma(\lambda)d\lambda} \qquad\qquad Eq\text{-}S5$$

where $I(\lambda)$ (photons $cm^{-2}$ $s^{-1}$) and $\sigma(\lambda)$ ($cm^2$ $molecules^{-1}$) are the actinic flux spectra of light source and the absorption cross section of the NPM, respectively (Figure R8).

In the revised SI, we have added Table R2 as Table S2, we added the discussion of quantum yields in line 230-232 in the revised manuscript. "In order to compare the efficiency of NPM at different $Fe^{3+}$ concentrations in producing HONO and NO, $\Phi_{HONO}$ and $\Phi_{NOx}$ are shown in Table S3. It can be concluded that photolysis of NPM in the presence of high $Fe^{3+}$ concentrations had more important contribution to HONO formation as compared to HONO level formed by photolysis of pure NPM. "

[Figure]

**Figure R8.** Absorption cross section of NPM at different $Fe^{3+}$ concentrations.

**Table R2.** Quantum Yields ($\phi$) for Photolysis of NPM and NOx at different $Fe^{3+}$ concentrations

| The concentration of $Fe^{3+}$ (mg ml$^{-1}$) | 0 | 0.1 | 0.25 | 0.5 | 0.8 |
|---|---|---|---|---|---|
| $\phi_{HONO}$ | $4.48\times10^{-5}$ | $1.84\times10^{-4}$ | $2.03\times10^{-4}$ | $1.59\times10^{-4}$ | $1.55\times10^{-4}$ |
| $\phi_{NO2}$ | $2.03\times10^{-5}$ | $6.46\times10^{-5}$ | $5.21\times10^{-5}$ | $4.71\times10^{-5}$ | $4.02\times10^{-5}$ |
| $\phi_{NO}$ | $4.06\times10^{-5}$ | $3.35\times10^{-5}$ | $2.10\times10^{-5}$ | $1.87\times10^{-5}$ | $1.40\times10^{-5}$ |

References:

Ahmed, S., Akhtar, N., Rahman, A., Mondal, N. C., Khurshid, S., Sarah, S., Muqtada Ali Khan, M.and Kamboj, V.: Evaluating groundwater pollution with emphasizing heavy metal hotspots in an urbanized alluvium watershed of Yamuna River, northern India, Environmental Nanotechnology, Monitoring & Management, 18, doi:10.1016/j.enmm.2022.100744, 2022.

Anderson, T. A., Salice, C. J., Erickson, R. A., McMurry, S. T., Cox, S. B.and Smith, L. M.: Effects of landuse and precipitation on pesticides and water quality in playa lakes of the southern high plains, Chemosphere, 92, 84-90, doi:10.1016/j.chemosphere.2013.02.054, 2013.

Aregahegn, K. Z., Ezell, M. J.and Finlayson-Pitts, B. J.: Photochemistry of Solid Films of the Neonicotinoid Nitenpyram, Environmental Science & Technology, 52, 2760-2767, doi:10.1021/acs.est.7b06011, 2018.

Aregahegn, K. Z., Shemesh, D., Gerber, R. B.and Finlayson-Pitts, B. J.: Photochemistry of Thin Solid Films of the Neonicotinoid Imidacloprid on Surfaces,

Environmental Science & Technology, 51, 2660-2668, doi:10.1021/acs.est.6b04842, 2017.

Batchelli, S., Muller, F. L. L., Chang, K.-C.and Lee, C.-L.: Evidence for Strong but Dynamic Iron−Humic Colloidal Associations in Humic-Rich Coastal Waters, Environmental Science & Technology, 44, 8485-8490, doi:10.1021/es101081c, 2010.

Bejan, I., Abd El Aal, Y., Barnes, I., Benter, T., Bohn, B., Wiesen, P.and Kleffmann, J.: The photolysis of -nitrophenols:: a new gas phase source of HONO, Physical Chemistry Chemical Physics, 8, 2028-2035, doi:10.1039/b516590c, 2006.

Bey, I., Jacob, D. J., Yantosca, R. M., Logan, J. A., Field, B. D., Fiore, A. M., Li, Q., Liu, H. Y., Mickley, L. J.and Schultz, M. G.: Global modeling of tropospheric chemistry with assimilated meteorology: Model description and evaluation, Journal of Geophysical Research: Atmospheres, 106, 23073-23095, doi:10.1029/2001jd000807, 2001.

Bonmatin, J. M., Giorio, C., Girolami, V., Goulson, D., Kreutzweiser, D. P., Krupke, C., Liess, M., Long, E., Marzaro, M., Mitchell, E. A. D., Noome, D. A., Simon-Delso, N.and Tapparo, A.: Environmental fate and exposure; neonicotinoids and fipronil, Environmental Science and Pollution Research, 22, 35-67, doi:10.1007/s11356-014-3332-7, 2014.

Borsuah, J. F., Messer, T. L., Snow, D. D., Comfort, S. D.and Mittelstet, A. R.: Literature Review: Global Neonicotinoid Insecticide Occurrence in Aquatic Environments, Water, 12, doi:10.3390/w12123388, 2020.

Brigante, M., Cazoir, D., D'Anna, B., George, C.and Donaldson, D. J.: Photoenhanced uptake of NO by pyrene solid films, Journal of Physical Chemistry A, 112, 9503-9508, doi:10.1021/jp802324g, 2008.

Cai, J., Zhi, G. R., Yu, Z. Q., Nie, P., Gligorovski, S., Zhang, Y. Z., Zhu, L. K., Guo, X. X., Li, P., He, T., He, Y. J., Sun, J. Z.and Zhang, Y.: Spectral changes induced by pH variation of aqueous extracts derived from biomass burning aerosols: Under dark and in presence of simulated sunlight irradiation, Atmospheric Environment, 185, 1-6, doi:10.1016/j.atmosenv.2018.04.037, 2018.

Cazoir, D., Brigante, M., Ammar, R., D'Anna, B.and George, C.: Heterogeneous photochemistry of gaseous NO on solid fluoranthene films: A source of gaseous nitrous acid (HONO) in the urban environment, Journal of Photochemistry and Photobiology a-Chemistry, 273, 23-28, doi:10.1016/j.jphotochem.2013.07.016, 2014.

Deguillaume, L., Leriche, M., Desboeufs, K., Mailhot, G., George, C.and Chaumerliac, N.: Transition metals in atmospheric liquid phases: Sources, reactivity, and sensitive parameters, Chemical Reviews, 105, 3388-3431, doi:10.1021/cr040649c, 2005.

Fukuhara, K., Kurihara, M.and Miyata, N.: Photochemical generation of nitric oxide from 6-nitrobenzo[a]pyrene, Journal of the American Chemical Society, 123, 8662-8666, doi:10.1021/ja0109038, 2001.

Gen, M., Zhang, R.and Chan, C. K.: Nitrite/Nitrous Acid Generation from the Reaction of Nitrate and Fe(II) Promoted by Photolysis of Iron–Organic Complexes, Environmental Science & Technology, 55, 15715-15723, doi:10.1021/acs.est.1c05641, 2021.

Gerringa, L. J. A., Rijkenberg, M. J. A., Wolterbeek, H. T., Verburg, T. G., Boye, M.and de Baar, H. J. W.: Kinetic study reveals weak Fe-binding ligand, which affects the solubility of Fe in the Scheldt estuary, Marine Chemistry, 103, 30-45, doi:10.1016/j.marchem.2006.06.002, 2007.

Gobler, C. J., Donat, J. R., Consolvo, J. A.and Sañudo-Wilhelmy, S. A.: Physicochemical speciation of iron during coastal algal blooms, Marine Chemistry, 77, 71-89, doi:Doi 10.1016/S0304-4203(01)00076-7, 2002.

González-Mariño, I., Rodríguez, I., Rojo, L.and Cela, R.: Photodegradation of nitenpyram under UV and solar radiation: Kinetics, transformation products identification and toxicity prediction, Science of the Total Environment, 644, 995-1005, doi:10.1016/j.scitotenv.2018.06.318, 2018.

Han, C., Yang, W. J., Wu, Q. Q., Yang, H.and Xue, X. X.: Key role of pH in the photochemical conversion of NO to HONO on humic acid, Atmospheric Environment, 142, 296-302, doi:10.1016/j.atmosenv.2016.07.053, 2016.

Hayat, W., Zhang, Y. Q., Hussain, I., Du, X. D., Du, M. M., Yao, C. H., Huang, S. B.and Si, F.: Efficient degradation of imidacloprid in water through iron activated sodium persulfate, Chemical Engineering Journal, 370, 1169-1180, doi:10.1016/j.cej.2019.03.261, 2019.

Hladik, M. L., Kolpin, D. W.and Kuivila, K. M.: Widespread occurrence of neonicotinoid insecticides in streams in a high corn and soybean producing region, USA, Environmental Pollution, 193, 189-196, doi:10.1016/j.envpol.2014.06.033, 2014.

Jiang, Y., Xue, L. K., Gu, R. R., Jia, M. W., Zhang, Y. N., Wen, L., Zheng, P. G., Chen, T. S., Li, H. Y., Shan, Y., Zhao, Y., Guo, Z. X., Bi, Y. J., Liu, H. D., Ding, A. J., Zhang, Q. Z.and Wang, W. X.: Sources of nitrous acid (HONO) in the upper boundary layer and lower free troposphere of the North China Plain: insights from the Mount Tai Observatory, Atmospheric Chemistry and Physics, 20, 12115-12131, doi:10.5194/acp-20-12115-2020, 2020.

Lacson, C. F. Z., de Luna, M. D. G., Dong, C. D., Garcia-Segura, S.and Lu, M. C.: Fluidized-bed Fenton treatment of imidacloprid: Optimization and degradation pathway, Sustainable Environment Research, 28, 309-314, doi:10.1016/j.serj.2018.09.001, 2018.

Lee, J. D., Whalley, L. K., Heard, D. E., Stone, D., Dunmore, R. E., Hamilton, J. F., Young, D. E., Allan, J. D., Laufs, S.and Kleffmann, J.: Detailed budget analysis of HONO in central London reveals a missing daytime source, Atmospheric Chemistry and Physics, 16, 2747-2764, doi:10.5194/acp-16-2747-2016, 2016.

Li, C. J., Zhang, D. H., Peng, J. L.and Li, X. G.: The effect of pH, nitrate, iron (III) and bicarbonate on photodegradation of oxytetracycline in aqueous solution, Journal of Photochemistry and Photobiology a-Chemistry, 356, 239-247, doi:10.1016/j.jphotochem.2018.01.004, 2018.

Lippiatt, S. M., Lohan, M. C.and Bruland, K. W.: The distribution of reactive iron in northern Gulf of Alaska coastal waters, Marine Chemistry, 121, 187-199, doi:10.1016/j.marchem.2010.04.007, 2010.

Lu, Z., Challis, J. K.and Wong, C. S.: Quantum Yields for Direct Photolysis of Neonicotinoid Insecticides in Water: Implications for Exposure to Nontarget Aquatic Organisms, Environmental Science & Technology Letters, 2, 188-192, doi:10.1021/acs.estlett.5b00136, 2015.

Malato, S., Caceres, J., Agüera, A., Mezcua, M., Hernando, D., Vial, J.and Fernández-Alba, A. R.: Degradation of imidacloprid in water by photo-fenton and TiO photocatalysis at a solar pilot plant:: A comparative study, Environmental Science & Technology, 35, 4359-4366, doi:10.1021/es000289k, 2001.

Monge, M. E., D'Anna, B., Mazri, L., Giroir-Fendler, A., Ammann, M., Donaldson, D. J.and George, C.: Light changes the atmospheric reactivity of soot, Proceedings of the National Academy of Sciences of the United States of America, 107, 6605-6609, doi:10.1073/pnas.0908341107, 2010.

Morrissey, C. A., Mineau, P., Devries, J. H., Sanchez-Bayo, F., Liess, M., Cavallaro, M. C.and Liber, K.: Neonicotinoid contamination of global surface waters and associated risk to aquatic invertebrates: A review, Environment International, 74, 291-303, doi:10.1016/j.envint.2014.10.024, 2015.

Nguyen, D. D. D., Huynh, K. A., Nguyen, X. H.and Nguyen, T. P.: Imidacloprid degradation by electro-Fenton process using composite Fe3O4–Mn3O4 nanoparticle catalyst, Research on Chemical Intermediates, 46, 4823-4840, doi:10.1007/s11164-020-04246-0, 2020.

Nishioka, J., Takeda, S., Wong, C. S.and Johnson, W. K.: Size-fractionated iron concentrations in the northeast Pacific Ocean: distribution of soluble and small colloidal iron, Marine Chemistry, 74, 157-179, doi:Doi 10.1016/S0304-4203(01)00013-5, 2001.

Öztürk, M., Bizsel, N.and Steinnes, E.: Iron speciation in eutrophic and oligotrophic Mediterranean coastal waters;: impact of phytoplankton and protozoan blooms on iron distribution, Marine Chemistry, 81, 19-36, doi:10.1016/S0304-4203(02)00137-8, 2003.

Palma, D., Arbid, Y., Sleiman, M., de Sainte-Claire, P.and Richard, C.: New Route to Toxic Nitro and Nitroso Products upon Irradiation of Micropollutant Mixtures Containing Imidacloprid: Role of NOx and Effect of Natural Organic Matter, Environmental Science & Technology, 54, 3325-3333, doi:10.1021/acs.est.9b07304, 2020.

Rózsa, G., Náfrádi, M., Alapi, T., Schrantz, K., Szabó, L., Wojnárovits, L., Takács, E.and Tungler, A.: Photocatalytic, photolytic and radiolytic elimination of imidacloprid from aqueous solution: Reaction mechanism, efficiency and economic considerations, Applied Catalysis B-Environmental, 250, 429-439, doi:10.1016/j.apcatb.2019.01.065, 2019.

Sedaghat, M., Vahid, B., Aber, S., Rasoulifard, M. H., Khataee, A.and Daneshvar, N.: Electrochemical and photo-assisted electrochemical treatment of the pesticide imidacloprid in aqueous solution by the Fenton process: effect of operational parameters, Research on Chemical Intermediates, 42, 855-868, doi:10.1007/s11164-015-2059-5, 2016.

Soltani-nezhad, F., Saljooqi, A., Shamspur, T.and Mostafavi, A.: Photocatalytic degradation of imidacloprid using GO/FeO/TiO-NiO under visible radiation: Optimization by response level method, Polyhedron, 165, 188-196, doi:10.1016/j.poly.2019.02.012, 2019.

Song, M., Zhao, X. X., Liu, P. F., Mu, J. C., He, G. Z., Zhang, C. L., Tong, S. R., Xue, C. Y., Zhao, X. J., Ge, M. F.and Mu, Y. J.: Atmospheric NO oxidation as major sources for nitrous acid (HONO), Npj Climate and Atmospheric Science, 6, doi:ARTN 3010.1038/s41612-023-00357-8, 2023.

Sun, Y. H.and Liu, X.: Efficient visible-light photocatalytic degradation of imidacloprid and acetamiprid using a modified carbon nitride/tungstophosphoric acid composite induced by a nucleophilic addition reaction, Applied Surface Science, 485, 423-431, doi:10.1016/j.apsusc.2019.04.203, 2019.

Wan, D., Sharma, V. K., Liu, L., Zuo, Y. G.and Chen, Y.: Mechanistic Insight into the Effect of Metal Ions on Photogeneration of Reactive Species from Dissolved Organic Matter, Environmental Science & Technology, 53, 5778-5786, doi:10.1021/acs.est.9b00538, 2019.

Wang, H., Lu, X., Jacob, D. J., Cooper, O. R., Chang, K.-L., Li, K., Gao, M., Liu, Y., Sheng, B., Wu, K., Wu, T., Zhang, J., Sauvage, B., Nédélec, P., Blot, R.and Fan, S.: Global tropospheric ozone trends, attributions, and radiative impacts in 1995–2017: an integrated analysis using aircraft (IAGOS) observations, ozonesonde, and multi-decadal chemical model simulations, Atmospheric Chemistry and Physics, 22, 13753-13782, doi:10.5194/acp-22-13753-2022, 2022.

Wang, W. C., Huang, D. Y., Wang, D. X., Tan, M. X., Geng, M. Y., Zhu, C. Y., Chen, N.and Zhou, D. M.: Extensive production of hydroxyl radicals during oxygenation of anoxic paddy soils: Implications to imidacloprid degradation, Chemosphere, 286, doi:ARTN 13156510.1016/j.chemosphere.2021.131565, 2022.

Wang, W. H., Aregahegn, K. Z., Andersen, S. T., Ni, A. Z., Rohrbacher, A. F., Nielsen, O. J.and Finlayson-Pitts, B. J.: Quantum Yields and NO Formation from Photolysis of Solid Films of Neonicotinoids, Journal of Agricultural and Food Chemistry, 67, 1638-1646, doi:10.1021/acs.jafc.8b05417, 2019.

Wang, W. H., Ezell, M. J., Lakey, P. S. J., Aregahegn, K. Z., Shiraiwa, M.and Finlayson-Pitts, B. J.: Unexpected formation of oxygen-free products and nitrous acid from the ozonolysis of the neonicotinoid nitenpyram, Proceedings of the National Academy of Sciences of the United States of America, 117, 11321-11327, doi:10.1073/pnas.2002397117, 2020.

Wang, Y. A., Fu, X., Wu, D. M., Wang, M. D., Lu, K. D., Mu, Y. J., Liu, Z. G., Zhang, Y. H.and Wang, T.: Agricultural Fertilization Aggravates Air Pollution by Stimulating Soil Nitrous Acid Emissions at High Soil Moisture, Environmental Science & Technology, 55, 14556-14566, doi:10.1021/acs.est.1c04134, 2021.

Yang, W., You, D., Li, C., Han, C., Tang, N., Yang, H.and Xue, X.: Photolysis of Nitroaromatic Compounds under Sunlight: A Possible Daytime Photochemical Source of Nitrous Acid?, Environmental Science & Technology Letters, 8, 747-752, doi:10.1021/acs.estlett.1c00614, 2021.

Yang, W. J., Yuan, H., Han, C., Yang, H.and Xue, X. X.: Photochemical emissions of HONO, NO and NO from the soil surface under simulated sunlight, Atmospheric Environment, 234, doi:ARTN 11759610.1016/j.atmosenv.2020.117596, 2020.

Zhou, S., Young, C. J., VandenBoer, T. C., Kowal, S. F.and Kahan, T. F.: Time-Resolved Measurements of Nitric Oxide, Nitrogen Dioxide, and Nitrous Acid in an Occupied New York Home, Environmental Science & Technology, 52, 8355-8364, doi:10.1021/acs.est.8b01792, 2018.

Zhou, W. T., Mekic, M., Liu, J. P., Loisel, G., Jin, B., Vione, D.and Gligorovski, S.: Ionic strength effects on the photochemical degradation of acetosyringone in atmospheric deliquescent aerosol particles, Atmospheric Environment, 198, 83-88, doi:10.1016/j.atmosenv.2018.10.047, 2019.

---

## Author Comment (AC2)

We appreciate the constructive comments of the reviewer on our manuscript. We have carefully responded to all comments accordingly. The revisions are described in details below.

**Response to Reviewer #2:**

This paper discussed one of the insecticide such as NPM and its aqueous photochemistry could be a significant source of HONO and NOx especially when only compared to soil emissions. If this is true, the widely usage of insecticide could be another reason for the general discover of the presence of HONO even in the rural and remote areas. This study therefore make an interesting contribution to the community to bridge the argriculature insecticide (kind of emerging pollutants) and the atmospheric chemistry. I suggest publication after the authors to address my following comments.

**General comments**

As the authors mentioned, the kinetic study is performed for extremely high concentration level (- 50000 imu g L^-1}) which is several orders of magnitude higher than the environmental concentrations. Would the kinetics be different for the much lower concentrations? The authors may add some uncertainty discussions on this aspect according to possible theoretical approaches.

**Response:** As the reviewer suggested, we explored the kinetics of NPM at lower concentration (0.1 mg $mL^{-1}$), which is close to the limit of detection considering both NPM and reactive nitrogen species detection. As shown in Figure R6 below and in Figure S6, the kinetic data has shown a robust linear relationship between NOx and HONO production and light density at different NPM levels. The obtained results show that the rate constant (k) is faster at low NPM concentration compared to that of high NPM concentrations. It is important to note that, although our kinetic experiments are not capable of simulating concentrations under the environmental concentration, we selected a rationalization parameter scheme related to the environmental concentration of NPM (50 $\mu$g $L^{-1}$) and soluble iron (92.48 nmol $L^{-1}$, 0.025 mg $L^{-1}$ in our study), in order to estimate the environmental NPM and iron concentration contributed to the formation of reactive nitrogen species which is representative to a certain level. In addition, the main scope of our study is to reveal that the light-induced degradation of NPM leads to enhanced production of HONO and NOx driven by secondary photochemistry between redox reaction of $Fe^{3+}/Fe^{2+}$ and photoproduced ROS. We quantified the photochemical HONO and NOx formation through NPM photodegradation, and we suggest that this chemistry may represent a significant source of HONO and NOx in the regions where surface waters are polluted with NNs insecticides.

In the revised manuscript, we added the sentence "The kinetic data has shown that the rate constant (k) is faster at low NPM concentration compared to that of high NPM concentrations (Figure S6)."

[Figure]

**Figure R6.** First-order rates at different NPM concentrations

**Specific comments**

1. HONO is indirectly measured as the difference between the $NO_2$ signal and the $Na_2CO_3$ tube. This could be subjected with some uncertainty. Other reactive nitrogen species might be included in this differential signal. The uncertainty discussions maybe added.

**Response:** NO, $NO_2$ and HONO concentrations were detected using a chemiluminescence NOx analyzer (42i, THERMO) with a molybdenum converter. Because HONO was detected by a quartz tube (25 cm length, 2.9 cm inner diameter) filled with $Na_2CO_3$ between the reactor and the analyzer was employed to remove HONO. The removal efficiency of HONO by the $Na_2CO_3$ tube reached 99% at the steady sate both in our study and previous work (Han et al., 2016). And the detection technique for HONO has been widely employed to measure the HONO concentration in many studies (Han et al., 2016; Yang et al., 2020). Meanwhile, it should be noted that $NO_2$ and NO are barely captured by the $Na_2CO_3$ tube as shown in the Figure R3. In order to validate the feasibility of the method, we also performed additional test experiment to confirm the indirectly determined HONO values by using Water-Based Long-Path Absorption Photometer (WLPAP, Beijing Zhichen Technology Co., Ltd, China) on-line connected with the reactor for real-time measurements of HONO, and the results agree well with the performed measurements by $Na_2CO_3$ tube (Figure R4).

[Figure]

**Figure R3.** The adsorption of $NO_2$ and NO in the reactor by $Na_2CO_3$ tube

[Figure]

**Figure R4.** Typical HONO profile measured in real time by WLPAP analyzer upon irradiation of NPM. Conditions: Irradiation intensity of 169.4 W m$^{-2}$ at 300< λ <400 nm, NPM concentration of 0.1 mg ml$^{-1}$, temperature of 298 K

2. Sect. 2.5, I have quickly checked the unit of the equation 3 and 4, I can end up with the unit of min^-1, but it is better converted to s^-1, the authors may further clarify it.

**Response:** Thank you for your suggestions and we apologize for the misunderstanding.

The photolysis frequencies (J-values, S$^{-1}$) of NPM to HONO and NOx were calculated by Eq.(3) and Eq.(4), respectively.

$$J_{NPM \to HONO} = \frac{QM_{NPM} \int_0^t C_t^{HONO} dt}{60 \times 10^{-3} N_A \times t \times (m_0 + m_t)/2} \tag{3}$$

$$J_{NPM \to NOx} = \frac{QM_{NPM} \int_0^t C_t^{NOx} dt}{60 \times 10^{-3} N_A \times t \times (m_0 + m_t)/2} \tag{4}$$

Where Q (mL min$^{-1}$) and M$_{NPM}$ (g mol$^{-1}$) are the total flow gas rates in the reactor and the molar mass of NPM, respectively; t (min) is the irradiation time; Ct$^{NOx}$ (molecules cm$^{-3}$) is the concentration of gaseous HONO or NOx formed by photolysis of NPM during the irradiation period; N$_A$ is the Avogadro number; M$_0$ (mg) and M$_t$ (mg) are the masses at the beginning and end of the NPM photolysis experiments.

The constant 1/60 in Eqs. 3 and 4 represents the conversion of min$^{-1}$ to S$^{-1}$.

3.  In the reference part, Wang Y et al., 2021 is duplicated.

**Response:** Thank you for noticing this. It is corrected.

References:

Han, C., Yang, W. J., Wu, Q. Q., Yang, H. and Xue, X. X.: Heterogeneous photochemical conversion of NO$_2$ to HONO on the humic acid surface under simulated sunlight. Environ. Sci. Technol. 2016, 50, 5017–5023.

Yang, W. J., Yuan, H., Han, C., Yang, H.and Xue, X. X.: Photochemical emissions of HONO, NO and NO from the soil surface under simulated sunlight, Atmospheric Environment. 2020, 234, 117596.